# Guided Diffusion by Optimized Loss Functions on Relaxed Parameters for Inverse Material Design

## Abstract

Inverse design problems are common in engineering and materials science. The forward direction, i.e., computing output quantities from design parameters, typically requires running a numerical simulation, such as a FEM, as an intermediate step, which is an optimization problem by itself. In many scenarios, several design parameters can lead to the same or similar output values. For such cases, multi-modal probabilistic approaches are advantageous to obtain diverse solutions. A major difficulty in inverse design stems from the structure of the design space, since discrete parameters or further constraints disallow the direct use of gradient-based optimization. To tackle this problem, we propose a novel inverse design method based on diffusion models. Our approach relaxes the original design space into a continuous grid representation, where gradients can be computed by implicit differentiation in the forward simulation. A diffusion model is trained on this relaxed parameter space in order to serve as a prior for plausible relaxed designs. Parameters are sampled by guided diffusion using gradients that are propagated from an objective function specified at inference time through the differentiable simulation. A design sample is obtained by backprojection into the original parameter space. We develop our approach for a composite material design problem where the forward process is modeled as a linear FEM problem. We evaluate the performance of our approach in finding designs that match a specified bulk modulus. We demonstrate that our method can propose multiple diverse designs within 1% relative error margin from medium to high target bulk moduli in 2D and 3D settings. We also demonstrate that the material density of generated samples can be minimized simultaneously by using a multi-objective loss function.

## 1 Introduction

Inverse design demands finding suitable design parameters for a requirement specified in terms of output quantities. For example, in composite material design, where multiple materials are combined at a microscopic scale, one might desire a specific property of the composite structure, such as a target bulk modulus, and then needs to find appropriate component materials and composition parameters. The bulk modulus characterizes a material's resistance to compression and is a key property in engineering applications. For example, an important inverse design problem is to identify parameters that realize a prescribed target bulk modulus while simultaneously minimizing material density. To evaluate the design parameters, a forward simulation such as a finite element method (FEM) is run, yielding output quantities which can then be fed into an objective function. The structure of the design space limits applicable methods, since for example non-differentiability of the simulation with respect to the design parameters disallows direct gradient-based approaches. Commonly, black-box optimization methods such as Bayesian optimization (BO, Frazier & Wang (2016)) or deep reinforcement learning (DRL, Würz & Weißenfels (2025)), which optimize an objective function by sequential trials, are employed. However, these approaches seek the optimal design instead of diverse designs which also achieve low objective cost.

In this paper, we propose a different, probabilistic approach for inverse design based on diffusion models. We consider a relaxation of the original design parameter space into a continuous grid representation, such as the individual elements of a FEM. This space is much higher-dimensional, but allows for differentiation through

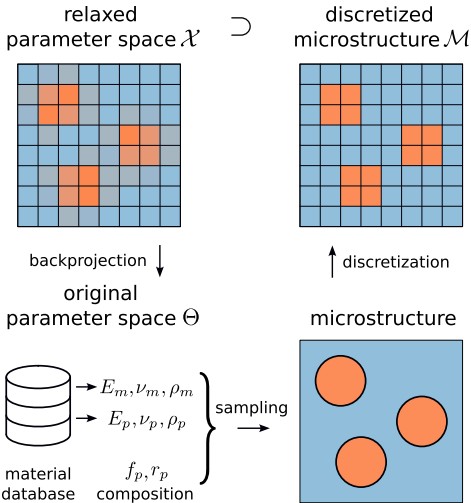

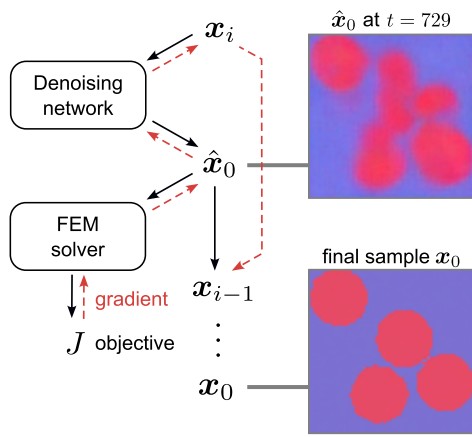

(a) Overview of the parameter spaces. A diffusion model on the relaxed parameter space $\mathcal{X}$ is trained to generate samples close to $\mathcal{M}$ (plausible discretized microstructures). The samples are then backprojected to the original parameter space.

(b) Overview of the guided sampling process. The denoising network prediction $\hat{x}_0$ is evaluated in an FEM solver. Implicit gradients of the objective function which depend on the FEM solution are propagated back to guide the denoising step.

Figure 1: Overview of our proposed method. Original design parameters are relaxed into a continuous grid representation on which a diffusion model acts as prior. We guide the sample generation process zero-shot by objective functions using implicit differentiation of the FEM solver. Our approach finds diverse samples with low objective cost.

the forward simulation, which is a constrained optimization problem itself, by implicit differentiation. Given differentiability of the objective function, one could apply gradient-based optimization in this space, but obtained samples would lose correspondence to the original parameters. We therefore train a diffusion model on the relaxed parameter space, which then acts as a prior for plausible relaxed designs that correspond to original parameters. We then perform guided diffusion (Chung et al., 2023; Song et al., 2023), which guides the generation of samples by gradients of a loss function towards samples that satisfy the desired criteria. Instead of using gradients of learned surrogate models of the objective (Yang et al., 2024), we directly employ gradients of the objective function propagated through the forward simulation, which constitutes an optimized loss function. The obtained samples are subsequently mapped back to the original design space by a domain-specific process. While components of our method such as loss-guided diffusion (Song et al., 2023), differentiating through FEM solvers (Xue et al., 2023) and the use of guided diffusion in material design (Yang et al., 2024; Zampini et al., 2025) have been proposed in previous work, we propose an approach for inverse design that combines them in a novel way. Our approach allows for obtaining design proposals in a parameter space for which the objective is not directly differentiable by relaxing the parametrization. While we evaluate our approach for a material design problem, in principle, it can be applied to similar problems which support differentiable relaxation of a non-differentiable design problem and optimized loss functions on the relaxed parameters.

We demonstrate our approach for a composite material design problem where the microstructure, i.e., the spatial representation of the composite structure at microscopic scale, is inhomogeneous due to the presence of spherical particles. To make numerical simulations feasible, the complex microstructure is represented by a smaller sample called a representative volume element (RVE). At the macroscale, this RVE is treated as if it were homogeneous, a concept known as homogenization. We consider isotropic, linear elastic properties for both matrix (the base or filling phase) and spherical included particles, which allows us to compute the bulk modulus from the stress response (Huet, 1990) by solving a linear FEM problem. The design space consists of the choice of base materials for matrix and particles as well as the particle volume fraction and radius. Base materials exist as discrete instances and the particle volume fraction and radius are not easily

differentiable due to the implicit constraint of integer numbers of particles. We relax the design problem by allowing arbitrary material properties in each element of the discretized microstructure, resulting in a pixel (2D) or voxel (3D) representation. We train a diffusion model on a dataset of plausible microstructures and then use it in conjunction with gradients computed by a FEM solver on the relaxed problem to optimize an objective function, while simultaneously staying on the manifold of plausible microstructures. Importantly, the diffusion model is independent of the objective function and can be reused for other targets, as long as we can compute the respective gradients. We evaluate our approach for 2D and 3D design problems for simulated materials where the goal is to achieve a prescribed macroscopic bulk modulus and also the minimization of density on 2D problems.

In summary, we contribute the following: (1) We propose a novel approach for inverse design based on diffusion models. We relax the original design space into a representation which can be used for evaluating optimization-based objective functions in a differentiable way. We guide the reverse diffusion by the objective function which enables flexible adaptation of the objective at inference time without retraining. (2) We develop and evaluate our approach for an inverse design problem of composite materials. Our results demonstrate that our approach can propose multiple diverse materials within a 5% relative error margin for all target bulk moduli and within 1% relative error margin for medium to high target bulk moduli. We also demonstrate that material density can be minimized simultaneously by a multi-objective loss without retraining the model.

## 2 Related Work

**Optimal Experimental Design.** Optimal Experimental Design methods (Franceschini & Macchietto, 2008) such as Bayesian optimization (BO (Foster et al., 2019; Frazier & Wang, 2016)) can be used to search for feasible design parameters for inverse design problems. Trials are conducted sequentially at promising parameters according to measures like information gain. BO uses the outcomes to update a regression function of the target value of an objective function. However, the regressor typically needs to be trained for each specific target value. Our diffusion model-based approach learns a prior over feasible designs in a relaxed space of the partially discrete design parameters. It generates diverse samples which are guided at inference time to achieve the design objective in a zero-shot manner.

**Guided Diffusion.** Several methods have been proposed that use diffusion models as priors and guide the reverse diffusion process by additional constraints. Diffusion Posterior Sampling (DPS, (Chung et al., 2023)) adds a Gaussian measurement to perform posterior inference. The diffusion is guided by the derivative of the squared residual between measurement and its expected value given the denoised state. The works in (Yu et al., 2023; Song et al., 2023) generalize this concept by energy- and loss-guided diffusion, respectively. In our approach, determining the expected measurement or loss requires solving an inner optimization problem. Some methods add proximal optimization steps to the reverse diffusion process (Song et al., 2022; Chung et al., 2024). Universal Guidance Diffusion (Bansal et al., 2023) proposes to learn a classifier function which is used to guide the diffusion process. In (Ye et al., 2024), several guidance approaches are unified in a single formulation. However, the above approaches do not consider inner optimization problems like our approach and cannot be directly applied to our inverse design problem without relaxing the parameter space.

**Inverse Material Design.** Previous work on the design of microstructures in the context of homogenization has primarily relied on evolutionary algorithms, such as Genetic Algorithms (GA) (Zohdi, 2003), or on deep reinforcement learning (DRL) approaches (Würz & Weißenfels, 2025). GAs have proven effective in tackling even non-convex optimization problems, like the case presented here. However, they are highly sensitive to initial conditions and often converge to local optima. Moreover, they struggle with handling complex and continuous state representations. The DRL-based method in (Würz & Weißenfels, 2025), on the other hand, explores the design space by trial-and-error to identify microstructures that exhibit desired effective material properties. The RL policy is trained for specific bulk modulus targets which is less flexible than our approach which allows for changing the objective at inference time. If the inverse design problem is directly differentiable for the parameters, gradient-based optimization can be employed (Xue et al., 2023). However, the design problem considered in this work has discrete parameters and is non-differentiable.

Diffusion approaches have been applied for related problems. In metamaterial design, the aim is to optimize small-scale structures which have a single material either added or removed at each location and give rise to

specific macroscopic properties. In contrast, our goal is to generate highly constrained microstructures which represent a discrete set of particles and instances from a discrete set of material properties. The approach in (Bastek & Kochmann, 2023) uses classifier-free guided diffusion to infer metamaterial shapes represented as 2D Gaussian random field. In (Liu et al., 2025), signed distance functions are instead used to model the metamaterial. Differently to our training-free method, the diffusion model needs to be trained on conditionals. Similar to our method, the preprint (Yang et al., 2024) proposes to use loss-guided diffusion for inverse design of metamaterials, but using a learned regressor of material properties. These works do not incorporate an FEM solver for guidance like our approach. In (Zampini et al., 2025), constraint projection is proposed for guided diffusion for metamaterial design. While the method uses FEM to determine the stress-strain curve, differently to our method, the solver is differentiated numerically using Monte-Carlo estimates for the guidance.

## 3 Preliminaries

### 3.1 Denoising Diffusion Probabilistic Models

In recent years, denoising diffusion probabilistic models (DDPM, (Ho et al., 2020)) have become a popular tool for image generation and inverse design. DDPM models a forward process that iteratively diffuses a data sample $\boldsymbol{x}_0$ in a latent variable $\boldsymbol{x}_t$ in successive time steps $t$. The forward diffusion process is modeled by a Gaussian distribution $q(\boldsymbol{x}_t \mid \boldsymbol{x}_0) = \mathcal{N}\left(\boldsymbol{x}_t; \sqrt{\bar{\alpha}_t}\boldsymbol{x}_0, (1 - \bar{\alpha}_t\boldsymbol{I})\right)$ conditional on the data sample, where $\bar{\alpha}_t := \prod_{s=1}^t \alpha_t$, $\alpha_t = 1 - \beta_t$, and $\beta_t$ are parameters of the variance schedule. We can write $\boldsymbol{x}_t(\boldsymbol{x}_0, \boldsymbol{\epsilon}) = \sqrt{\bar{\alpha}_t}\boldsymbol{x}_0 + \sqrt{1 - \bar{\alpha}_t}\boldsymbol{\epsilon}$ for $\boldsymbol{\epsilon} \sim \mathcal{N}(\boldsymbol{0}, \boldsymbol{I})$ in terms of $\boldsymbol{x}_0$ and $\boldsymbol{\epsilon} = \boldsymbol{\epsilon}(\boldsymbol{x}_t, \boldsymbol{x}_0)$. For $T \to \infty$, the latent variable tends to $\boldsymbol{x}_T \sim \mathcal{N}(\boldsymbol{0}, \boldsymbol{I})$. Ho et al. (2020) showed that this diffusion process can be reverted by iteratively sampling a latent variable for the previous time step, where a neural network with parameters $\theta$ predicts the noise $\boldsymbol{\epsilon}_\theta(\boldsymbol{x}_t, t)$. It is trained on the forward diffusion noise $\boldsymbol{\epsilon}(\boldsymbol{x}_t, \boldsymbol{x}_0)$. Song et al. (2020) further showed that this reverse process can be generalized, allowing an arbitrary number of sampling steps $N$, where then each $i \in \{0, N-1\}$ corresponds to a certain timestep $t$ from a subset of the original $T - 1$ to 0 timesteps. Note that $\boldsymbol{\epsilon}_\theta(\boldsymbol{x}_t, t)$ is closely related to the score of $p_t(\boldsymbol{x}_t) = \int q(\boldsymbol{x}_t \mid \boldsymbol{x}_0)p(\boldsymbol{x}_0)d\boldsymbol{x}_0$ and it holds $\nabla_{\boldsymbol{x}_t} \log p_t(\boldsymbol{x}_t) \approx -\frac{1}{\sqrt{1-\bar{\alpha}_t}}\boldsymbol{\epsilon}_\theta(\boldsymbol{x}_t, t)$ (Luo, 2022). There exist equivalent formulations for the noise $\boldsymbol{\epsilon}$, such as the "predicted clean sample" $\hat{\boldsymbol{x}}_0 = \frac{1}{\sqrt{\bar{\alpha}_t}}\boldsymbol{x}_t - \sqrt{1 - \bar{\alpha}_t}\boldsymbol{\epsilon}$. An alternative is the velocity parameterization proposed in (Salimans & Ho, 2022): $\boldsymbol{v} = \sqrt{\bar{\alpha}_t}\boldsymbol{\epsilon} - \sqrt{1 - \bar{\alpha}_t}\hat{\boldsymbol{x}}_0$. All of the three quantities can be used as targets for the neural network. During denoising, they can be converted from one to another as shown. We train our network to predict the quantity $\boldsymbol{v}$ as $\boldsymbol{v}_\theta$ and convert this to $\hat{\boldsymbol{x}}_0 = \sqrt{\bar{\alpha}_t}\boldsymbol{x}_t - \sqrt{1 - \bar{\alpha}_t}\boldsymbol{v}_\theta$ for denoising.

We apply DDIM (Song et al., 2020) to reduce the number of iterations in the reverse diffusion process. A sample from the previous DDIM distribution $x_{i-1}$ can be obtained as

$$\boldsymbol{x}_{i-1} = \frac{\sqrt{\tilde{\alpha}_i}(1 - \bar{\alpha}_{i-1})}{1 - \bar{\alpha}_i}\boldsymbol{x}_i + \frac{\sqrt{\bar{\alpha}_{i-1}}\tilde{\beta}_i}{1 - \bar{\alpha}_i}\hat{\boldsymbol{x}}_0 + \sigma_i\boldsymbol{z} \tag{1}$$

where $\boldsymbol{z} \sim \mathcal{N}(\boldsymbol{0}, \boldsymbol{I})$, $\tilde{\alpha}_i = \bar{\alpha}_i/\bar{\alpha}_{i-1}$, $\tilde{\beta}_i = 1 - \tilde{\alpha}_i$ and $\sigma_i = \sqrt{(1 - \bar{\alpha}_{i-1})/(1 - \bar{\alpha}_i)\tilde{\beta}_i}$, and $i$ is the DDIM iteration.

### 3.2 Loss-Guided Diffusion

Using Bayes' rule, diffusion posterior sampling (DPS (Chung et al., 2023)) combines the prior learned by the diffusion model with likelihoods of additional measurements $\boldsymbol{y}$, $\nabla_{\boldsymbol{x}_t} \log p_t(\boldsymbol{x}_t \mid \boldsymbol{y}) = \nabla_{\boldsymbol{x}_t} \log p_t(\boldsymbol{x}_t) + \nabla_{\boldsymbol{x}_t} \log p_t(\boldsymbol{y} \mid \boldsymbol{x}_t)$. DPS proposes to approximate the intractable likelihood $p_t(\boldsymbol{y} \mid \boldsymbol{x}_t)$ with $p_t(\boldsymbol{y} \mid \hat{\boldsymbol{x}}_0)$. Loss-guided diffusion (Song et al., 2023) extends this approach to arbitrary loss functions $\ell_{\boldsymbol{y}}(\hat{\boldsymbol{x}}_0)$ by choosing $p(\boldsymbol{y}|\hat{\boldsymbol{x}}_0) = \frac{1}{Z}\exp(-\ell_{\boldsymbol{y}}(\hat{\boldsymbol{x}}_0))$, where $Z$ is the partition function. This allows for approximating

$$\nabla_{\boldsymbol{x}_t} \log p_t(\boldsymbol{y} \mid \boldsymbol{x}_t) \approx \nabla_{\boldsymbol{x}_t} \log p_t(\boldsymbol{y} \mid \hat{\boldsymbol{x}}_0) = -\nabla_{\boldsymbol{x}_t}\ell_{\boldsymbol{y}}(\hat{\boldsymbol{x}}_0). \tag{2}$$

To implement the guidance, we follow DPS and add $-\rho_D\nabla_{\boldsymbol{x}_i}\ell_{\boldsymbol{y}}(\hat{\boldsymbol{x}}_0)$ to $x_{i-1}$ in the denoising step, where $\rho_D$ is a scaling parameter.

## 4 Guided Diffusion with Optimized Loss Functions

### 4.1 Optimized Loss Functions for Inverse Design

We consider discretized inverse design problems, with parameters $\boldsymbol{\theta} \in \Theta$ for which the quality of the design is evaluated using an objective function $J(\boldsymbol{\theta}, \boldsymbol{u})$. $J$ depends on the design parameters $\boldsymbol{\theta}$ and also on the solution $\boldsymbol{u}(\boldsymbol{\theta}) \in \mathbb{R}^N$ of a forward optimization problem that depends on the design $\boldsymbol{\theta}$. For example, one can calculate the inner displacements in a microstructure with FEM given the design parameters. The solution $\boldsymbol{u}$ needs to satisfy some constraint function $c(\boldsymbol{\theta}, \boldsymbol{u}) = \mathbf{0}$. The goal is to minimize the objective function $J(\boldsymbol{\theta}, \boldsymbol{u})$, i.e., $\min_{\boldsymbol{\theta} \in \Theta, \boldsymbol{u} \in \mathbb{R}^N} J(\boldsymbol{\theta}, \boldsymbol{u})$ s.t. $c(\boldsymbol{\theta}, \boldsymbol{u}) = \mathbf{0}$. Note that these constraints are not the constraints on the designs. For many problems, $J$ is not differentiable w.r.t. to $\boldsymbol{\theta}$. For example, in our composite material design problem, the parameters for base material properties need to be chosen from a discrete set of available materials and the parameters volume fraction and particle radius need to be represented by an integer number of particles for calculating $J$ using FEM simulation. We relax the parameter space into another continuous parameter space $\mathcal{X} = \mathbb{R}^K$, which is potentially of much higher dimensionality than the original $\Theta$. Valid relaxed parameterizations $\boldsymbol{m} \in \mathcal{M}$ that correspond to possible original design parameters form a subset $\mathcal{M}$ and for each $\boldsymbol{\theta} \in \Theta$, there exists a $\boldsymbol{m} \in \mathcal{M}$. We further assume that $J$ can be expressed in $\boldsymbol{x}$ as $J(\boldsymbol{x}, \boldsymbol{u}(\boldsymbol{x}))$ and that $J$ is differentiable w.r.t. to $\boldsymbol{u}$ and $\boldsymbol{x}$. An example of $\boldsymbol{m}$, as used in our experiments, is a 2D or 3D discretization into finite elements of a microstructure consisting of a matrix and particles. The $\boldsymbol{x}$ are parameter grids with arbitrary material properties of individual pixels or voxels at each element. In the relaxed parameter space, the implicit function theorem can be used to determine the total differential of $J$ w.r.t. $\boldsymbol{x}$ (Xue et al., 2023), i.e.

$$\frac{\mathrm{d}J}{\mathrm{d}\boldsymbol{x}} = -\frac{\partial J}{\partial \boldsymbol{u}} \left( \frac{\partial c}{\partial \boldsymbol{u}} \right)^{-1} \frac{\partial c}{\partial \boldsymbol{x}} + \frac{\partial J}{\partial \boldsymbol{x}} \qquad \text{by using} \qquad \frac{\mathrm{d}\boldsymbol{u}}{\mathrm{d}\boldsymbol{x}} = -\left( \frac{\partial c}{\partial \boldsymbol{u}} \right)^{-1} \frac{\partial c}{\partial \boldsymbol{x}}. \tag{3}$$

Note that the presented formulation might be impractical for computation and one can instead use an adjoint formulation (Strang, 2007). Also note that optimizing the objective function using gradient descent is not sufficient due to the high-dimensional ambiguous parameter space and missing constraints of the original design space (e.g., discrete number and spherical particles, discrete set of materials) which requires suitable means for regularization.

### 4.2 Regularization by Guided Diffusion

Instead of solving the inverse design problem in the original design space, we propose to find possible solutions in the relaxed parameter space. To approximate the constraints of the design problem, we regularize gradient-based optimization in the relaxed space using a diffusion model of valid relaxed parameters as prior. We first train an unconditional diffusion model on a training set of plausible designs which are in $\mathcal{M}$. We then use loss guidance to infer relaxed parameters that remain close to the training data manifold and minimize the loss function $\ell_{\boldsymbol{y}}(\hat{\boldsymbol{x}}_0) = J(\boldsymbol{y}, \hat{\boldsymbol{y}}(\hat{\boldsymbol{x}}_0, \boldsymbol{u}))$. The loss can be chosen, for instance, as the squared error between the expected measurement $\hat{\boldsymbol{y}}(\hat{\boldsymbol{x}}_0, \boldsymbol{u})$ and its target value $\boldsymbol{y}$, i.e., $J(\boldsymbol{y}, \hat{\boldsymbol{y}}(\hat{\boldsymbol{x}}_0)) = \|\boldsymbol{y} - \hat{\boldsymbol{y}}(\hat{\boldsymbol{x}}_0)\|_2^2$. The expected measurement is calculated from parameters $\hat{\boldsymbol{x}}_0$ and the solution $\boldsymbol{u}$ for the constraint function $c$. For example, we can choose $\boldsymbol{y}$ as a target bulk modulus of a material, and $\hat{\boldsymbol{y}}(\hat{\boldsymbol{x}}_0, \boldsymbol{u})$ as the bulk modulus of the material generated by guided diffusion. The final generated sample $\boldsymbol{x}_0$ is then backprojected to the original design space by a domain-specific process, yielding the proposed design parameter $\boldsymbol{\theta}$.

## 5 Matrix-Particle Inverse Material Design by Loss-Guided Diffusion

We consider an inverse material design problem in which circular or spherical particles, respectively, of a certain radius and material are mixed into a base material called the matrix with a certain volume fraction. Our goal is to infer the parameters of the particles and the matrix materials in a microstructure of specific square 2D or cubic 3D size. Following the idea of homogenization, the macroscopic, averaged physical properties are calculated on the microstructure (Temizer & Zohdi, 2007). This requires a specific set of

possible loadings (i.e., displacements at the boundary) on the microstructure to fulfill energetic requirements (Hill, 1972). To determine macroscopic properties, either the stress or the strain must be constant on average over the microstructure and independent of the distribution and composition of the materials. In this study, the constant strain approach is applied. In addition, we impose displacements linearly to all points of the surface to fulfill (Hill, 1972). In the case of isotropy, the material can be described using two parameters, such as Young's modulus $E$ and Poisson ratio $\nu$ (Zohdi & Wriggers, 2008). Additionally, we consider the density $\rho$ of the material. This leads to parameters $(E_m, \nu_m, \rho_m)$ for the matrix, in which particles of common radius $r_p$ with material properties $(E_p, \nu_p, \rho_p)$ are mixed. The total volume fraction of particles is denoted by $f_p$ which is implemented by an integer number of particles in our generated microstructures. Our design parameters are therefore $\boldsymbol{\theta} = (E_m, E_p, \nu_m, \nu_p, \rho_m, \rho_p, r_p, f_p)$ and we are mainly interested in achieving a specific macroscopic bulk modulus $K$. The averaged, homogenized $K$ is calculated from the average of the stress over the entire volume and the previously defined strain. $K = \frac{\text{tr}\langle\boldsymbol{\sigma}\rangle}{3\,\text{tr}\,\boldsymbol{\varepsilon}}$ where $\boldsymbol{\varepsilon}$ denotes the prescribed strain and $\langle\boldsymbol{\sigma}\rangle$ the stress averaged over the microstructure. Boundary conditions are applied as $\overline{\boldsymbol{u}}(\boldsymbol{q}) = \boldsymbol{\varepsilon}\boldsymbol{q}$ where $\overline{\boldsymbol{u}}(\boldsymbol{q})$ denotes the displacement at the surface point of the microstructure with position vector $\boldsymbol{q}$ (Zohdi & Wriggers, 2008).

## 5.1 Discrete Implementation

The finite element method (FEM) is used to compute the stress distribution within the microstructure (Zohdi & Wriggers, 2008) for calculating its bulk modulus. For this purpose, the volume is subdivided into equally sized elements. The adjacent connection points of the elements are referred to as nodes. The linear displacement is imposed on all boundary nodes on the surface. A discretized microstructure $\boldsymbol{m} \in \mathcal{M}$ consists of such an element grid where each element is assigned to either matrix or particle material. Particles are circles resp. spheres and equally sized. We additionally assume that there is an integer number of particles and they do not intersect the boundaries. Consequently, one can calculate design parameters $\boldsymbol{\theta}$ from $\boldsymbol{m}$, but not all $\boldsymbol{\theta}$ can directly be represented as a single microstructure. To determine reliable macroscopic measures, several such microstructures need to be sampled and their results averaged to assess the homogenization. Solutions for the FEM can be obtained by solving a linear system $\boldsymbol{A}\boldsymbol{u} = \boldsymbol{b}$. Note that $\frac{\partial c}{\partial \boldsymbol{u}} = \boldsymbol{A}$ and since $\boldsymbol{A}$ is symmetric for this problem, the left term of Equation (3) can be obtained as $\boldsymbol{p} = \frac{\partial J}{\partial \boldsymbol{u}}\boldsymbol{A}^{-1}$ by solving the system $\boldsymbol{A}^\top \boldsymbol{p}^\top = \boldsymbol{A}\boldsymbol{p}^\top = \left(\frac{\partial J}{\partial \boldsymbol{u}}\right)^\top$. Here, $\boldsymbol{A}$ depends on the microstructure $\boldsymbol{m}$ which itself depends on $\boldsymbol{\theta}$. Our first considered objective function is measuring the squared error of the predicted material's $K$ to a specific prescribed $K^*$, i.e., $J_1(K, K^*) = (K - K^*)^2$. Our second considered objective function incorporates $J_1$ and additionally the density of the microstructure, i.e., $J_2(K, K^*, \rho_m, \rho_p, f_p) = (K - K^*)^2 + \lambda\left((1 - f_p)\,\rho_m + f_p\,\rho_p\right)$, thus the goal is to match a prescribed $K^*$ while also minimizing density.

## 5.2 Guided Diffusion of Material Parameters

While these objective functions are differentiable w.r.t. the material properties $(E_m, E_p, \nu_m, \nu_p, \rho_m, \rho_p)$, the spatial configuration and the radius and number of particles for a volume fraction are highly ambiguous. Moreover, optimizing the discrete number of particles in valid spatial configurations is challenging. Also, the possible set of materials is constrained to a specific discrete set of usable known materials (e.g., specific types of rubber, metals). Without proper regularization, this optimization problem is ill-posed. We use diffusion models to learn a prior over valid relaxed material parameters. We use it in guided diffusion to perform the regularized optimization.

To parametrize materials in a continuous form which is suited for diffusion models, we represent materials as finite element discretizations $\boldsymbol{x} \in \mathcal{X}$ similar to $\mathcal{M}$, but each element $e$ can have arbitrary material properties $(E^e, \nu^e, \rho^e)$. The relaxed material parameters which we optimize by guided diffusion are thus the element materials in the finite element grid. This representation is 2D-image- or 3D-grid-like and can be embedded into a latent representation using a common U-Net (Ronneberger et al., 2015) architecture. We train the diffusion model with samples that are in $\mathcal{M}$ to learn a prior over valid microstructures in the relaxed parameter space. To generate an example, we randomly sample the properties of matrix and particle from a table of possible materials. We also sample volume fraction $f_p$ and particle radius $r_p$ randomly, determine the number of particles by rounding the quotient of $f_p$ and the area $\pi r_p^2$ resp. volume $\frac{4}{3}r_p^3$ and create a microstructure $\boldsymbol{m}$ with randomized, non-overlapping particle positions.

### 5.3 Backprojection

Given an objective function, e.g., to achieve a bulk modulus, our guided diffusion yields parameters $\boldsymbol{x}$, for which we now need to find the most accurate match $\hat{\boldsymbol{\theta}} \in \Theta$. Ideally, we expect the resulting sample to have a structure where particles can be distinguished from the matrix, all particles have the same radius and the materials for all particles and for all matrix pixels, respectively, are identical. Compare Figure 2 for an exemplary set of samples. In principle, a domain expert can analyze the result and determine validity and estimate a matching $\hat{\boldsymbol{\theta}}$. However, we describe an automatic approach that we apply in our experiments. First, we fit a 2-component Gaussian mixture model on the vectorized parameter image or grid on the 3-channel material data $E, \nu, \rho$. For this model, we prescribe spherical covariances for simplicity. We expect two peaks, if the composite material consists of different materials for particles and matrix, or one, if there are no particles or they have the same material. The means are the estimates for the materials $\hat{E}_1, \hat{E}_2, \hat{\nu}_1, \hat{\nu}_2, \hat{\rho}_1, \hat{\rho}_2$. We sum up the variances of all components as metric $V_m$ to asses how clearly the generated materials can be distinguished. For example, for unconditional generation this variance is 0.0011 on average (see Table 5). Refer to appendix section C for more details.

Next, we try to distinguish elements into matrix or particle material. Each element is assigned to the closest of both detected materials. We test both possibilities of assigning a material to the particles and identify circles (2D) or spheres (3D) and their median radius based on skeletonization (Lee et al., 1994). We choose the assignment with lowest variance in detected radii. This yields $r_p$ as mean of matched radii and we obtain the volume fraction $f_p$ by counting the number of particle and matrix material assignments to elements. Note that this does not necessarily result in an integer number of particles. Details of our algorithm can be found in appendix section C. In the list of available materials, we search for the nearest neighbor of our predicted materials for matrix and particle. We measure distance in a normalized space $[-1, 1]^3$ as metric. The nearest existing materials together with $r_p$ and $f_p$ form our prediction $\hat{\boldsymbol{\theta}}$ in the original parameter space $\Theta$. To approximate the expected value of the bulk modulus for such $\hat{\boldsymbol{\theta}}$, we sample several microstructures with discrete number of particles and average the results.

## 6 Experiments

### 6.1 Experiment Setup

We omit units throughout the paper and specify $E$ and $K$ in GPa and $\rho$ in $\mathrm{g\,cm^{-3}}$. For our experiments, we selected properties $(E, \nu, \rho)$ of 500 materials according to the online database MatWeb [1] with properties $0 < E \leq 500, 0 < \nu < 0.5, 0 < \rho < 10$. Since more than 8000 materials fulfill these properties and their distribution density varies greatly (e.g. there exist hundreds of entries for steel with similar properties), we slice the three property dimensions into 10 equidistant segments each and obtain 1000 chunks. We query the database for each of the individual chunks and subsample retrieved materials per chunk so that the total number does not exceed 500, a limitation required by MatWeb's terms of use. To this end, we compute a maximum number of materials that any chunk can contain and subsample accordingly. 168 of the chunks are nonempty. We show the distribution of the materials in appendix section A. We normalize each dimension between $[-1, 1]$, which constitutes the space in which our model is trained and our distances for nearest neighbor-matching, as well as variances for material matching are reported. To obtain a balanced training set, we first sample one of the nonempty chunks and then uniformly sample one of the available materials inside that chunk for both matrix and particles. We then sample volume fraction and particle radius as detailed in appendix section A. We then sample the particle positions (see Section 5) and obtain a discretized microstructure for diffusion and FEM. For our training dataset, we sample 10,000 examples. We evaluate our approach at several target values $K^*$ for both objective functions $J_1$ and $J_2$. We determine those as equally spaced values in an interval between the 1 and 99-percentiles of $K$ values on a set of 10k samples created similarly to our training set. See appendix section A for a histogram of $K$ values. We tune guidance parameters on the start (1-percentile), end, and midpoint values (for 2D: 4.8, 168.5, 332.2) and additionally report results on the 25% and 75% positions in the interval (86.6, 250.4).

---

[1] https://www.matweb.com/

### 6.2 Denoising, Training, and Evaluation Metrics

**Model and training**  For implementing our model and diffusion process, we use the Diffusers library (von Platen et al., 2022). The architecture is based on UNet (Ronneberger et al., 2015) using either 2D or 3D convolutions. The model employs ResNet layers and two downsampling steps, each halving the input dimensions. Before the respective upsampling stage, two more ResNet layers with self attention layers are employed. We detail the model architecture in appendix section D. For training, we use a cosine learning schedule with peak learning rate $10^{-3}$ and 5,000 warmup steps. We use the AdamW optimizer (Loshchilov & Hutter, 2019) with PyTorch default parameters $\beta_1 = 0.9, \beta_2 = 0.999, \epsilon = 10^{-8}, \lambda = 10^{-2}$. We clip the gradient norm at 1. We train our models for 100,000 steps with a batch size of 128.

**Denoising**  We use DDIM with $\eta = 1$ and by default $N = 100$ denoising time steps. We use a linear $\beta$-schedule between $\beta_0 = 10^{-5}$ and $\beta_T = 10^{-2}$ and employ $\beta$-rescaling as described by (Lin et al., 2024) which ensures that no information of the clean sample is left at $t = T$. We sample time steps that correspond to their "trailing" strategy. After denoising, we clip the sample to lie in the boundaries $[-1, 1]$. We use DPS with constant scaling parameter $\rho_D = 1$ for guidance in our experiments. Gradients from the FEM solver are scaled according to the derivative of the normalization since our diffusion model operates on a normalized space. We clip the gradient norms at 2.5 by rescaling gradients so that their norm does not exceed this value. Additionally, during guidance, we clip the predicted $\hat{x}_0$ to ensure it does not exceed boundaries. Please refer to the appendix section E for more details on the choice of guidance parameters.

**Evaluation metrics**  For assessing how well our approach finds parameters that achieve a desired $K^*$, we generate samples with guided diffusion and perform backprojection which yields a set of valid materials and particle parameters. To estimate the bulk modulus corresponding to these extracted parameters, similarly to our dataset generation, we generate multiple microstructures with random spatial particle distribution, compute their bulk modulus via FEM and average over them, yielding the quantity $K_\theta$. Details can be found in appendix section B. For the qualitative experiments, we additionally compute the bulk modulus of the particular generated sample and report it as $K_s$. We consider the relative error $\epsilon_r = |K_\theta - K^*|/K^*$ as primary metric. However, since we propose a probabilistic method that generates random samples by diffusion, it is interesting to know how many samples fall within some relative or absolute error margin. Therefore, for a set of generated samples with same target $K^*$, we compute how many of the samples have, e.g., $\epsilon_r < 1\%$ or absolute error $\epsilon = |K_\theta - K^*| < 5$ and denote this fraction by *frac*. For small sample sets, this quantity has a high variance, which can be reduced by obtaining more samples. It is also important to assess the diversity of generated samples, for which problem-specific metrics are required. For example, FID (Heusel et al., 2017) is commonly used in natural image generation (Dhariwal & Nichol, 2021) and several distance-based diversity metrics have been proposed for 3D point cloud generation (Yang et al., 2019). We propose two new diversity metrics for our specific material design problem. Firstly, to measure diversity of used base materials, the metric *cov* states how many of the nonempty base material chunks were matched by a material after backprojection of samples inside some error margin. Secondly, to directly analyze the diversity of generated design parameters after backprojection, the metric *ent* computes the mean entropy over multiple-resolution discretizations of the parameter space $\Theta$ of samples within the error margin. More details can be found in appendix section B.

### 6.3 Results

We first evaluate our approach on models trained on 10k samples for a 2D problem discretized into $64 \times 64$ elements. In the first two rows of Table 1, we assess the quality of 200 samples obtained for the $J_1$ objective function for each target $K^*$ (averaged over 3 model training seeds). We observe that more samples satisfy the error margins at medium $K^*$ values. We note that the samples exhibit a diverse coverage of material chunks, as, e.g., for $K^* = 168.5$, the average 115 samples that satisfy $\epsilon_r < 5\%$ cover 83 of the 168 material chunks on average. Also the ent metrics are always bigger than zero and indicate that diverse parameters are found, with highest value at at $K^* = 168.5$. For the very small target $K^* = 4.8$, the relative error margins become very tight in absolute terms. We present an evaluation with absolute error margins in the last rows of Table 1 and see that for e.g. $K^* = 4.8$ still 11.3% of samples fall into the margin $\epsilon < 1$.

Table 1: Evaluation of our method, projection variant and alternative inverse design approaches on the 2D problem on objective $J_1$. Metrics are computed over 200 samples each at different target $K^*$ and averaged over 3 training seeds for each method. $K^*$ marked with $'$ are part of the targets used for (guidance) hyper parameter optimization. $\dagger$: trained for specific objective; $\ddagger$: trained for specific target values.

| $K^*$ | | frac | | | | | cov | | | | | ent | | | | |
|---|---|---|---|---|---|---|---|---|---|---|---|---|---|---|---|---|
| | | $4.8'$ | $86.6$ | $168.5'$ | $250.4$ | $332.2'$ | $4.8'$ | $86.6$ | $168.5'$ | $250.4$ | $332.2'$ | $4.8'$ | $86.6$ | $168.5'$ | $250.4$ | $332.2'$ |
| ours | $\epsilon_r < 1\%$ | 0.012 | 0.030 | 0.147 | 0.100 | 0.055 | 0.026 | 0.054 | 0.230 | 0.081 | 0.056 | 1.000 | 1.928 | 4.711 | 3.774 | 3.056 |
| | $\epsilon_r < 5\%$ | 0.030 | 0.187 | 0.575 | 0.498 | 0.255 | 0.056 | 0.202 | 0.496 | 0.218 | 0.139 | 2.362 | 4.820 | 6.322 | 5.593 | 4.639 |
| ours, | $\epsilon_r < 1\%$ | 0.008 | 0.058 | 0.233 | 0.235 | 0.038 | 0.018 | 0.093 | 0.284 | 0.157 | 0.044 | 0.667 | 3.187 | 5.206 | 4.801 | 2.399 |
| proj. | $\epsilon_r < 5\%$ | 0.038 | 0.335 | 0.867 | 0.683 | 0.250 | 0.063 | 0.254 | 0.492 | 0.224 | 0.165 | 2.701 | 5.261 | 6.766 | 5.700 | 4.645 |
| cond. | $\epsilon_r < 1\%$ | 0.007 | 0.058 | 0.130 | 0.120 | 0.062 | 0.016 | 0.121 | 0.214 | 0.129 | 0.081 | 0.667 | 3.500 | 4.551 | 4.238 | 3.385 |
| diff$^\dagger$ | $\epsilon_r < 5\%$ | 0.042 | 0.335 | 0.570 | 0.515 | 0.285 | 0.079 | 0.427 | 0.456 | 0.300 | 0.262 | 2.984 | 5.796 | 6.434 | 6.100 | 5.450 |
| BO$^{\dagger\ddagger}$ | $\epsilon_r < 1\%$ | 0.002 | 0.083 | 0.072 | 0.123 | 0.150 | 0.004 | 0.089 | 0.117 | 0.109 | 0.117 | 0.000 | 3.703 | 3.620 | 4.103 | 4.357 |
| | $\epsilon_r < 5\%$ | 0.003 | 0.487 | 0.393 | 0.617 | 0.623 | 0.008 | 0.268 | 0.318 | 0.248 | 0.282 | 0.000 | 5.733 | 5.730 | 6.083 | 5.963 |
| RL$^{\dagger\ddagger}$ | $\epsilon_r < 1\%$ | 0.000 | 0.097 | 0.175 | 0.173 | 0.087 | 0.000 | 0.117 | 0.177 | 0.151 | 0.087 | 0.000 | 3.972 | 4.844 | 4.803 | 3.774 |
| | $\epsilon_r < 5\%$ | 0.002 | 0.213 | 0.272 | 0.333 | 0.182 | 0.004 | 0.190 | 0.240 | 0.218 | 0.159 | 0.000 | 4.988 | 5.436 | 5.616 | 4.806 |
| | $\epsilon < 1$ | 0.113 | 0.032 | 0.082 | 0.033 | 0.018 | 0.135 | 0.058 | 0.151 | 0.042 | 0.024 | 4.084 | 2.262 | 3.934 | 2.376 | 1.731 |
| ours | $\epsilon < 5$ | 0.653 | 0.225 | 0.393 | 0.185 | 0.085 | 0.274 | 0.230 | 0.417 | 0.115 | 0.075 | 5.533 | 5.057 | 5.891 | 4.351 | 3.545 |
| | $\epsilon < 10$ | 0.860 | 0.438 | 0.660 | 0.415 | 0.160 | 0.323 | 0.337 | 0.524 | 0.196 | 0.105 | 5.933 | 5.893 | 6.481 | 5.358 | 4.103 |

For the various absolute error margins, the fraction of samples within the margin decreases for higher targets after $K^* = 168.5$. We hypothesize that lower performance for higher targets stems from properties of the dataset (see appendix section A for details): Few microstructures in the training set have high $K$ values and also the gaps between observed $K$ become wider. Additionally, we observe that the gaps between $K$ values of the base materials become much larger with increasing $K$, which means that the result after backprojection can potentially deviate more strongly. We perform a more detailed analysis of the effect of backpropagation in appendix section C. We also show results for a variant "proj." that uses $N = 200$ timesteps, but alternates between a guidance step and a step that performs part of the backprojection on $\hat{x}_0$, replacing material values with the nearest neighbors of actually available materials. We observe that indeed frac metrics are considerably improved, whereas diversity metrics are slightly improved in most cases. We conduct subsequent experiments on the variant without projection as default due to the increased computational cost with projection.

We show the best samples from a single trained model in terms of $\epsilon_r$ and for higher quantiles of $\epsilon_r$ for selected targets $K^*$ in Figure 2 and for additional targets in appendix section G. One can see that, e.g., for $K^* = 168.5$, many diverse designs are generated, as can be seen by the varying shapes and colors. Even for higher error quantiles of that target, one can observe that the generated sample still matches the target well ($K_s$ similar to $K^*$). For the worst sample (100% quantile), the result after the backprojection ($K_\theta$) deviates stronger in most cases. One possibility is that the model generated materials that are not available or the generated structure could not be well matched by the backprojection. An interesting case is the best sample for $K^* = 250.4$, where the model generated an elongated particle. Still, the parameters extracted after backprojection lead to a similar result as this particular sample. The worst example of $K^* = 332.2$ demonstrates an implausible design with artifacts.

**Variants** In Table 2a, we compare the performance of our default models with models that are trained on only 1k samples. We see that the latter perform worse only by a small margin, hinting that fewer training samples could be used than in our default setting. We also compare our default denoising time steps of 100 against 50 and 200 and observe that both yield relatively similar results, where some metrics are slightly better and some slightly worse than our default setting. Consequently, denoising with $N = 50$ timesteps (meaning also only 49 gradient computations) could be a viable speedup, since the time depends linearly on $N$. For example, obtaining 200 samples with $N = 100$ took approximately 1.5h on a cluster node using

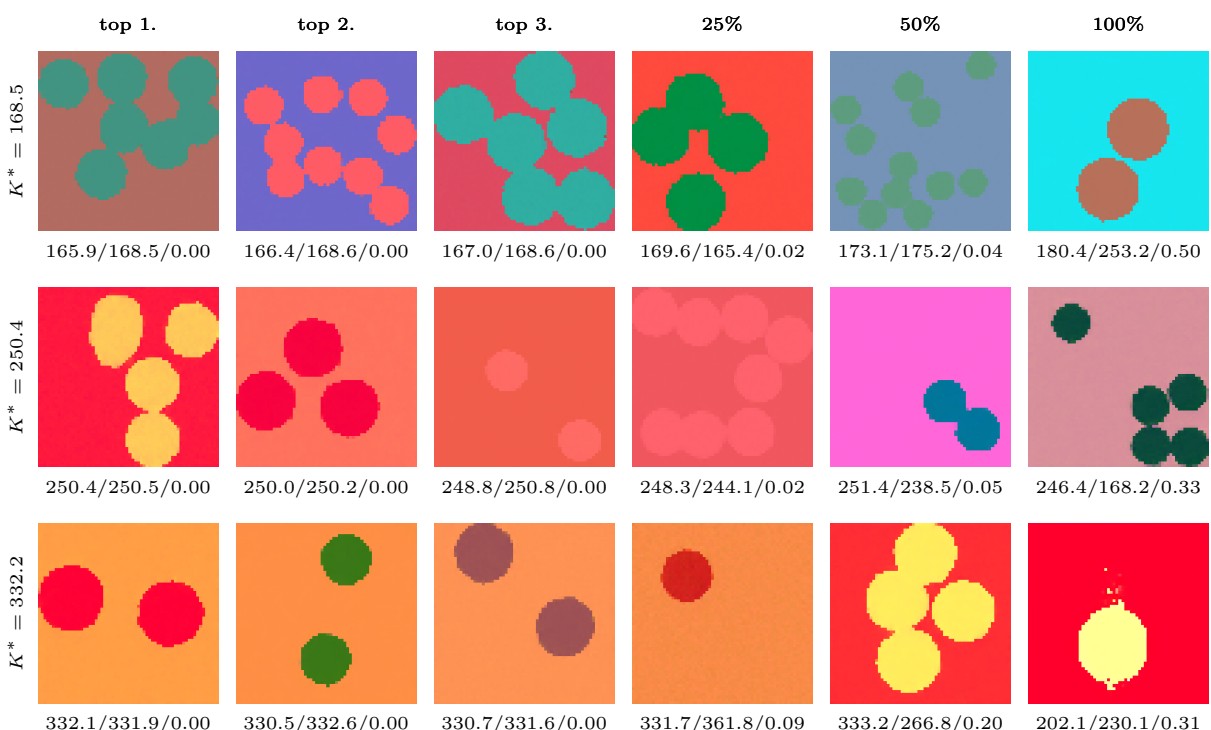

Figure 2: Inverse 2D material designs. Generated samples for selected bulk moduli $K^*$, ordered by relative error quantile. Best on the left, worst on the right. Labels show $K_s$ / $K_\theta$ / $\epsilon_r$. The values $(E, \nu, \rho)$ in the normalized coordinate space are encoded as $(r, g, b)$ values of the image. Our model is able to propose diverse and plausible designs close to the target bulk moduli.

Table 2: Performance for guidance by $J_1, J_2$ with ablations. Mean metrics over all targets $K^*$ for 200 samples each and over 3 model training seeds.

(a) Performance for guidance by $J_1$, variants.

|  | $\epsilon_r < 1\%$ | | | $\epsilon_r < 5\%$ | | |
|---|---|---|---|---|---|---|
|  | frac | cov | ent | frac | cov | ent |
| ds 10k | 0.069 | 0.089 | 2.894 | 0.309 | 0.222 | 4.747 |
| ds 1k | 0.052 | 0.077 | 2.368 | 0.259 | 0.183 | 4.006 |
| $N = 50$ | 0.060 | 0.081 | 2.600 | 0.299 | 0.225 | 4.711 |
| $N = 200$ | 0.066 | 0.073 | 2.830 | 0.317 | 0.201 | 4.653 |
| unguided | 0.002 | 0.004 | 0.067 | 0.019 | 0.037 | 1.342 |
| project | 0.115 | 0.119 | 3.252 | 0.435 | 0.240 | 5.015 |

(b) Performance for guidance by $J_2$ (also minimize density). Additionally we show the average density of samples that fall in the respective error margin.

|  | $\epsilon_r < 1\%$ | | | | $\epsilon_r < 5\%$ | | | |
|---|---|---|---|---|---|---|---|---|
|  | $\rho_{\text{avg}}$ | frac | cov | ent | $\rho_{\text{avg}}$ | frac | cov | ent |
| $\lambda = 0$ | 4.382 | 0.069 | 0.089 | 2.894 | 4.312 | 0.309 | 0.222 | 4.747 |
| $\lambda = 10^{-4}$ | 3.869 | 0.074 | 0.088 | 2.939 | 3.952 | 0.310 | 0.214 | 4.696 |
| $\lambda = 10^{-3}$ | 2.945 | 0.036 | 0.047 | 1.839 | 2.611 | 0.204 | 0.140 | 3.460 |
| $\lambda = 10^{-2}$ | 1.609 | 0.019 | 0.020 | 1.246 | 1.513 | 0.108 | 0.051 | 2.655 |

16 CPU cores and an A40 GPU. Further details on runtime are found in the appendix section G. We also provide a baseline that uses no guidance at all which demonstrates that guidance strongly improves sample adherence to the objective.

**Multiple objectives** We also evaluate our method on a different objective function, $J_2$, which incorporates minimizing the average density of the sample as additional objective. Results in Table 2b for varying factors of the density penalty term show that indeed with stronger penalty, the average densities are reduced further. This also leads to lower performance metrics, which is expected since the set of acceptable samples is reduced. This experiment demonstrates how our approach can be used to adapt the objective function in a zero-shot way without retraining the model.

Table 3: Evaluation of our method on the 3D problem on objective $J_1$. Metrics are computed over 200 samples each at different target $K^*$ and averaged over 3 model training seeds.

| | metric: frac | | | | | cov | | | | | ent | | | | |
| --- | --- | --- | --- | --- | --- | --- | --- | --- | --- | --- | --- | --- | --- | --- | --- |
| $K^*$: | 0.9 | 83.3 | 165.6 | 248.0 | 330.3 | 0.9 | 83.3 | 165.6 | 248.0 | 330.3 | 0.9 | 83.3 | 165.6 | 248.0 | 330.3 |
| $\epsilon_r < 1\%$ | 0.003 | 0.050 | 0.118 | 0.188 | 0.047 | 0.008 | 0.097 | 0.179 | 0.093 | 0.036 | 0.000 | 3.199 | 4.402 | 4.006 | 1.862 |
| $\epsilon_r < 5\%$ | 0.020 | 0.240 | 0.510 | 0.608 | 0.305 | 0.040 | 0.308 | 0.357 | 0.224 | 0.093 | 1.947 | 5.260 | 6.153 | 5.516 | 4.022 |
| $\epsilon < 1$ | 0.483 | 0.057 | 0.072 | 0.077 | 0.013 | 0.308 | 0.109 | 0.127 | 0.056 | 0.018 | 5.789 | 3.348 | 3.717 | 3.170 | 1.032 |
| $\epsilon < 5$ | 0.792 | 0.285 | 0.337 | 0.328 | 0.093 | 0.387 | 0.337 | 0.308 | 0.139 | 0.044 | 6.354 | 5.472 | 5.666 | 4.664 | 2.624 |
| $\epsilon < 10$ | 0.937 | 0.518 | 0.593 | 0.510 | 0.188 | 0.419 | 0.409 | 0.383 | 0.202 | 0.071 | 6.546 | 6.240 | 6.330 | 5.311 | 3.417 |

**Comparison to alternative methods** We compare our approach to established methods for inverse material design methods in the 2D setting regarding matching a prescribed target $K^*$. Details can be found in the appendix section F. Firstly, we compare to a Bayesian optimization (BO) approach which models a surrogate cost function for a certain target $K^*$ using Gaussian processes. We perform the optimization process for objective function $J_1$ with the same targets $K^*$ as used to evaluate our method with 1000 steps each. Results are presented in Table 1. For $K^* = 4.8$ and 168.5, our approach outperforms dedicated BOs in all metrics and error margins. For the largest target bulk modulus, a higher fraction and more diverse samples can be found with the Bayesian optimization approach. We note, however, that for each of the target bulk moduli (or generally, different objectives), the whole BO process, including exploration of the space and fitting the surrogate function, has to be performed from scratch. To obtain a sample, sampling and minimization of the surrogate function has to be performed. Our method in contrast can be applied directly to new objectives. Additionally, we compare to an approach using reinforcement learning. The setting is similar to the Bayesian optimization considered above, but instead of modeling a surrogate cost function, neural networks are trained to directly propose designs maximizing a reward. Each policy is trained to achieve a specific target bulk modulus. We consider a multi-step setting where the agent receives feedback on its proposal and may adjust its search accordingly. Our approach outperforms RL in all targets but $K^* = 86.6$ in frac $\epsilon_r < 5\%$, wheres RL is stronger in the $\epsilon_r < 1\%$ margin in most targets. Regarding the cov metric, our model performs better or equally well in all but the highest target for $\epsilon_r < 5\%$ and at $K^* = 4.8$ and 168.5 for $\epsilon_r < 1\%$. We note that differently to our method but similarly to BO, training has to be performed for each target $K^*$ from scratch.

We also compare our approach to a setting where the diffusion model is trained on a dataset with annotated bulk moduli $K$ as conditional input. This allows to perform sampling with classifier-free guidance (Ho & Salimans, 2022) as for example employed by Yang et al. (2024). While this approach does not optimize an objective function, the goal of matching a prescribed target $K^*$ is similar to our objective $J_1$. We observe that the conditional diffusion method can achieve higher metrics than our method in several cases. Our approach beats conditional diffusion on all metrics for $\epsilon_r < 1\%$ at target bulk moduli 4.8 and 168.5 and has only slightly lower frac metrics for the two highest targets. In most cases, the sample diversity is higher with conditional diffusion. We note, however, that conditional diffusion models require an explicitly provided label to condition on during training and inference time. Objectives like $J_2$ which penalizes density without an explicit density target cannot be implemented in this setting without further guidance mechanisms. Also, a modification of the type of conditional is not possible without retraining, while our approach is more general and can be adapted to various objective functions in a zero-shot way.

**3D problem.** We also train models on 10k samples of a 32x32x32 discretized 3D problem and perform guided sampling with the same parameters as in 2D for $J_1$. Results for different target bulk moduli $K^*$ are shown in Table 3. A challenging target is $K^* = 0.9$ as it is extremely low. still, 48.3% of samples fall in the absolute error margin $\epsilon < 1$. We provide visualizations of 3D samples generated by a single trained model in Figure 3 and for additional targets in appendix section G. Our model proposes diverse and plausible designs, even for high error quantiles.

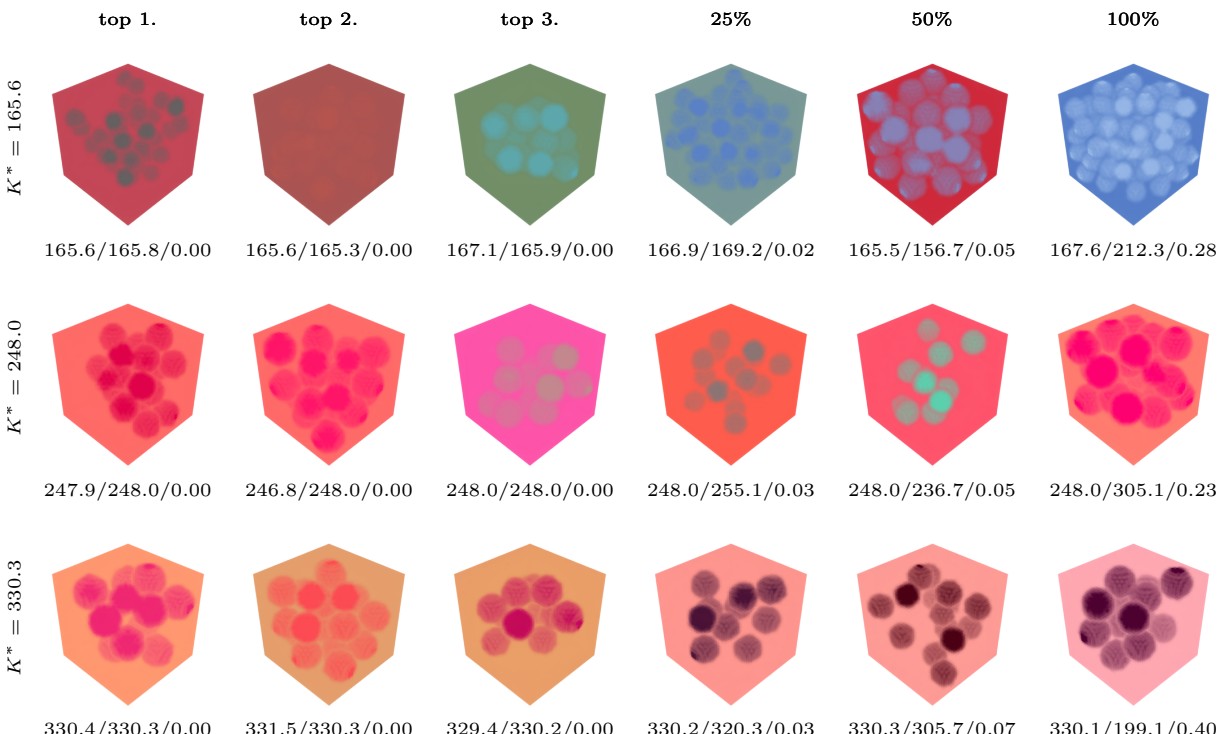

Figure 3: Inverse 3D material designs. Generated samples for selected bulk moduli $K^*$, ordered by relative error quantile. Best on the left, worst on the right. Labels show $K_s \, / \, K_\theta \, / \, \epsilon_r$. The values $(E, \nu, \rho)$ in the normalized coordinate space are encoded as $(r, g, b)$ values of the image. Our model is able to propose diverse and plausible designs close to the target bulk moduli.

## 7  Conclusion

In this paper, we develop a novel approach for inverse design based on loss-guided diffusion in which the loss is evaluated by solving an inner optimization problem. We evaluate our method for an inverse material design problem which requires solving a linear FEM to assess the bulk modulus of composite material microstructures. The microstructure consists of spherical particles in a matrix. Our approach operates on a relaxed reparameterization of the original parameter space which allows for denoising 2D and 3D grid representations of the microstructures. The diffusion model acts as prior and approximates the constraints of the design problem. Approximate design samples are projected back into the original design space. Our approach can directly leverage physics-based simulation to determine the loss function and does not require training a surrogate model for the loss.

We evaluate our method using a dataset of real material properties and demonstrate that our approach finds multiple diverse samples within a relative error of 1% from medium to high target bulk moduli in 2D and 3D settings. Our approach can optimize multiple objectives, which we demonstrate by also minimizing the density of generated samples in addition to matching a specified bulk modulus.

We anticipate that our approach will inspire future research in guided diffusion with optimization-based loss functions for inverse design in various application areas. In this work, we only considered linear FEMs as inner optimization problems. Future work could also evaluate and extend the method for non-linear optimization problems if implicit differentiation is possible. Another interesting direction of future research is to investigate implementing parallel solvers on GPU or incremental solvers which can reuse solutions of the inner optimization problem from previous diffusion iterations to improve runtime efficiency, especially for larger-scale or non-linear problems. From the material design perspective, investigating non-linear material properties and anisotropic materials is an interesting avenue of future research.

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

# Appendix

## A    Material List and Dataset Details

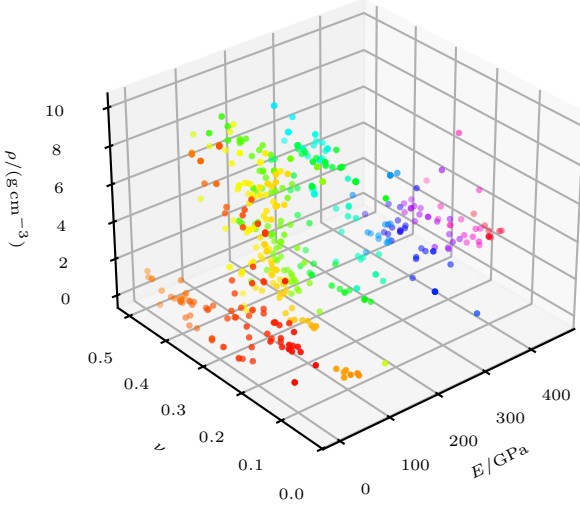

Figure 4: Visualization of base materials used. Color represents the index in the 168 non-empty chunks.

**Material list**    For our experiments, we selected properties $(E, \nu, \rho)$ of 500 materials from the online database MatWeb [2]. The original data is subject to copyright and terms of use of MatWeb. Due to license terms, our derived datasets and models cannot be made publicly available. Figure 4 shows the distribution of the base materials we used. The value ranges are $E \in [0.0055, 462], \nu \in [0.032, 0.499], \rho \in [0.032, 9.99]$.

**Dataset sampling**    When sampling an example for our datasets, we proceed in the following way: For both materials of the matrix and the particles, we first uniformly sample a non-empty chunk (out of 168) and then uniformly sample a base material that is contained in that chunk. The same distribution is used for matrix and particle material. After sampling the volume fraction $v_f$ and particle radius $r_p$, the number of particles is determined by dividing the volume fraction by the area (2D) or volume (3D) of a circle resp. sphere of that radius and rounding the result. For 2D, we sample $v_f$ uniformly in $[0.05, 0.5]$ and $2 \cdot r_p$ (diameter) uniformly in $[0.15, 0.4]$ where the unit refers to the relative size to the microstructure. For 3D, the ranges for $v_f$ are $[0.05, 0.45]$ and for $2 \cdot r_p$ (diameter) $[0.15, 0.35]$. Note that there are limits to those quantities, since circles or spheres need to be packed tightly, which becomes hard or even impossible with random sampling for higher volume fractions. This can result in a slightly different volume fraction than initially sampled. Note that we consider a unit square resp. cube.

To determine non-overlapping sphere positions (likewise for circles), we first randomly sample positions for all spheres so that boundaries are not intersected. We then compute the distances between all spheres and return if there are no intersections. If there are intersections, we determine the directions between all sphere centers and add small random vectors to better resolve penetrations. For each pairwise intersection between spheres, the delta in the direction that would resolve this intersection is added times 1.5 to a total delta per sphere position. Then, these deltas are applied at once for all spheres and the intersection check is done again. We stop this iterative resolving after 10,000 unsuccessful updates and re-sample initial positions in that case.

After obtaining an example microstructure, we can determine its bulk modulus $K$ with a FEM solver. For simpler computation, we use a 3D FEM solver also for our 2D case, where we build a thin 3D plane with a thickness of one element in one dimension. This represents plane stress conditions (the structure can move in this dimension, but the stress is zero). Note that this is, however, not necessary to train the diffusion model.

---

[2] https://www.matweb.com/

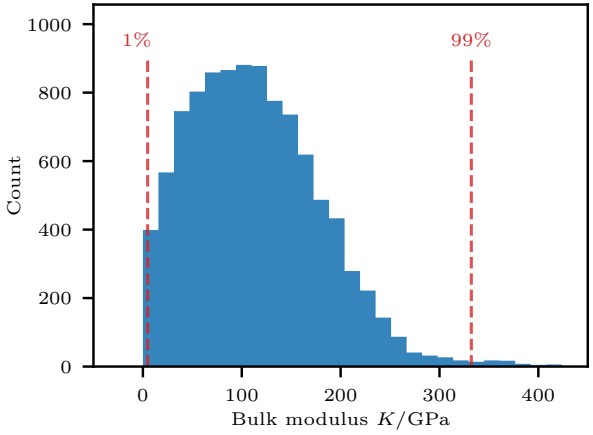 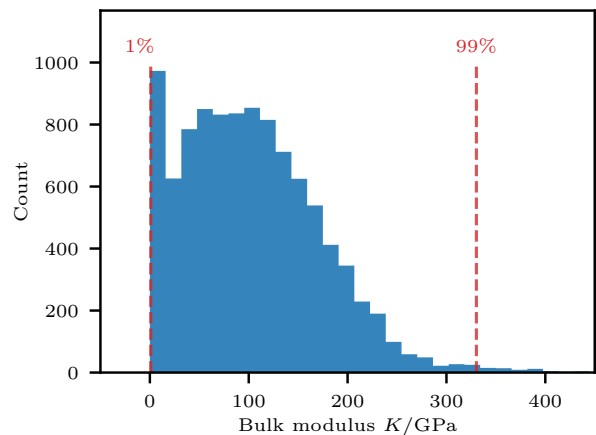

(a) K histogram (50 bins) of 2D problem ($64 \times 64$), otherwise unused seed. Cut at $K = 450$, maximum 784.5.

(b) K histogram (50 bins) of 3D problem ($32 \times 32 \times 32$), otherwise unused seed. Cut at $K = 450$, maximum 795.4.

Figure 5: Histograms of bulk modulus $K$ of individual examples in datasets. Guidance targets are chosen uniformly spaced between the 1 and 99 percentile.

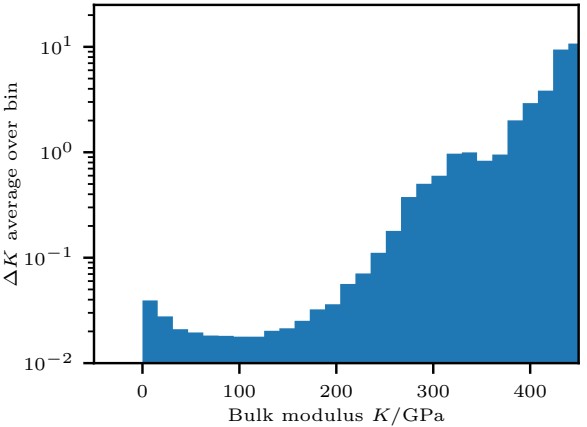 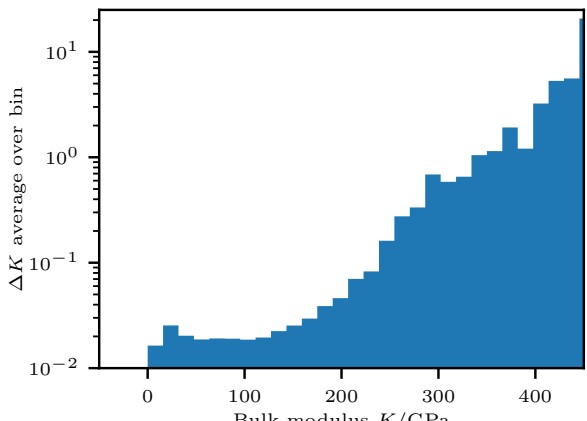

(a) Difference $\Delta K$ of sample's $K$ to next higher value, averaged over $K$ bin (50 bins total). 2D problem ($64 \times 64$). Logarithmic $y$ scale, cut at $K = 450$.

(b) Difference $\Delta K$ of sample's $K$ to next higher value, averaged over $K$ bin (50 bins total). 3D problem ($32 \times 32 \times 32$). Logarithmic $y$ scale, cut at $K = 450$.

Figure 6: Bin averages of $\Delta K$ (difference to next higher value) of individual examples in datasets.

**Analysis of $K$ values in datasets**   As described in section Section 6.1, we create datasets similar to our training datasets for the purpose of finding suitable target bulk moduli $K^*$. We show histograms of the bulk moduli of those datasets in Figure 5. We choose the range of target bulk moduli $K^*$ for evaluation as the 1% and 99% quantiles from these datasets.

We also analyze the gaps in $K$ values between samples in the dataset. To this end, we use the same datasets and histogram bins as for quantile computation. We sort all samples according to their $K$ value and compute the differences $\Delta K$ to the next higher value. We then average this difference per bin and show the results in Figure 6. One can observe that the gaps strongly increase from approximately $K = 150$ to $K = 450$, both for the 2D and 3D problmes. We perform a similar evaluation on the list of base materials and show the results in Figure 7.

We validate the resulting bulk modulus of samples for the 3D problem by computing the Voigt-Reuss bounds (Zohdi & Wriggers, 2008) from the material of matrix and particle for 500 samples generated similarly

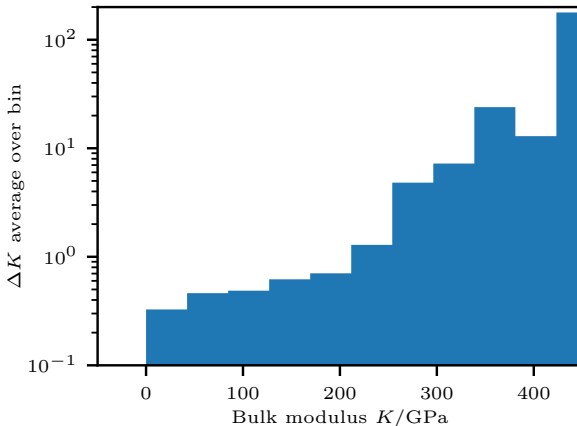

Figure 7: Difference $\Delta K$ of base material's $K$ to next higher value, averaged over $K$ bin (20 bins total). Logarithmic $y$ scale, cut at $K = 450$.

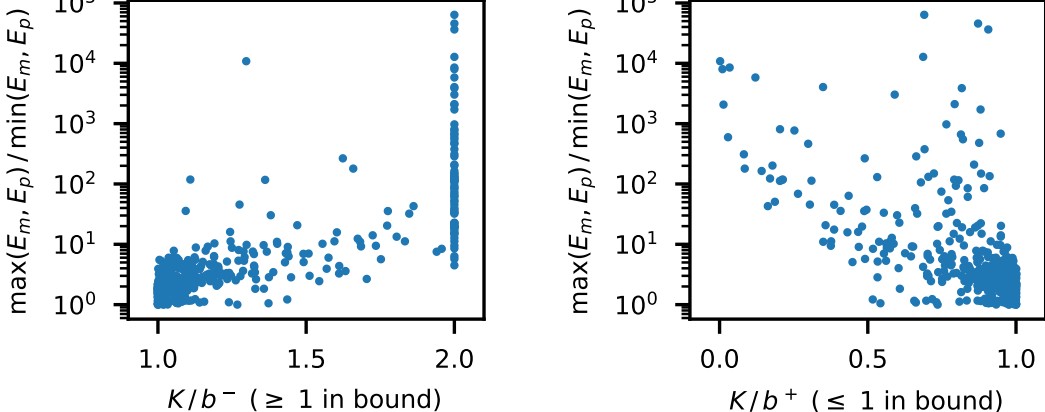

Figure 8: Relation between quotient of Elastic modulus of matrix and particle (y-axis) and quotient of bulk modulus K and lower ($b^-$) / upper ($b^+$) Voigt-Reuss bounds (Zohdi & Wriggers, 2008). Shown are 500 samples generated similarly to our 3D training dataset. The lower bound fraction is clipped at 2 for better display.

to our 3D training dataset and show the results in Figure 8. Out of the 500 samples, 33 were outside the bounds, but only with minor deviation, as evident from the plot. Due to the allocation of materials to the entire element, sawtooth-shaped transitions occur. We hypothesize that slightly excessive stresses may occur at the edges that can lead to values slightly above the bounds.

## B    Evaluation Metric Details

To determine the resulting bulk modulus for parameters $\boldsymbol{\theta}$ obtained from backprojection, we generate $n = 10$ microstructures with random spatial distribution of particles, similar to our dataset generation. Since we require an integer number of discrete particles for the sampled microstructure, we sample this number to accurately represent the predicted volume fraction (e.g., if the required number of particles computed by volume fraction and diameter is 3.2, we use 3 circles with probability 0.8 and 4 with 0.2). The bulk modulus is computed again via FEM and averaged over the $n$ samples, yielding the quantity $K_\theta$.

**cov metric**    To determine the cov metric, we consider the nearest actually available materials as computed by the backprojection. Due to the high variability in the base material sample density, we count the number of unique base material chunks instead of the individual materials. This requires the model to generate samples that yield nearest neighbors in every part of the material space. The cov metric computes the quotient of these chunks occurring in the generated sample set divided by the number of nonempty chunks. Note that this metric is highly dependent on the number of samples.

**ent metric**    To determine the ent metric, we divide each dimension of the parameter space $\Theta$ into equally sized bins, where the bin count equal the number 2 raised to varying powers. This leads to a finer granularity per exponent. Per exponent and per sample set inside an error margin, we compute the entropy in bits over unique discretizations of found parameters. For example, an entropy of 1 would correspond to only two different existing discretizations. We calibrate the highest used exponent of 2 on our whole training set and find that the entropy does not increase significantly after $2^6$ bins with a value of 13.29. The metric *ent* is computed by taking the mean over such entropies with exponents reaching from 1 to 6.

## C    Backprojection Details and Evaluation

In the following, we detail how we identify particles in the microstructure using the GMMs fitted to the material channels. For each element, we obtain a binary label indicating which component of the GMMs it belongs to with highest probability. This allows to represent the microstructure as a binary mask. The next step requires determining which label corresponds to the matrix and which to particles. We try to match circles resp. spheres for both cases of treating either the one- or zero-labels as foreground pixels, where the foreground is the candidate for the particles. To this end, we obtain the 2D resp. 3D skeleton by skeletonization (Lee et al., 1994) and their minimum distance to background pixels. We use some domain knowledge and filter out points with too small distances to the boundary of the segment or skip the whole case if there are too large distances. We also skip the case if more than half of the edge pixels resp. face voxels are positives, which should not be possible since we consider problems where particles do not intersect with the boundary. For all remaining foreground pixels $i$ in the skeleton and their minimum distance to background $d_i$, we check if there is another foreground pixel $j$ in the skeleton within $d_i$ that itself has $d_j < d_i$. In this case, we remove $j$. Afterwards, we are left with likely centers of circles resp. spheres and can use their distances as radius. If both cases have not been skipped, we compute the variance of remaining distances and choose the case with lower variance.

**Additional metrics**    To assess the quality of (potentially unconditionally) generated samples themselves, without comparing to an external objective, we employ the following metrics: Firstly, the sum of the variances of material fitting $V_m$: During the backprojection, we fit a 2-component GMM to the 3-channel material data. We sum up the fitted variances of both components to form our metric $V_m$. In the ideal case, this variance is 0, meaning that only 1 or 2 values ever occur as material values. This measures how well a model can enforce consistency of the materials in a sample.

| $K^*$ | 4.8 | 86.6 | 168.5 | 250.4 | 332.2 |
|---|---|---|---|---|---|
| $V_m$ | 0.044 | 0.002 | 0.001 | 0.009 | 0.008 |
| $d_m$ | 0.118 | 0.144 | 0.113 | 0.224 | 0.283 |
| frac $\epsilon_r < 1\%$ *sample* | 0.055 | 0.048 | 0.298 | 0.692 | 0.677 |
| frac $\epsilon_r < 5\%$ *sample* | 0.150 | 0.572 | 0.985 | 0.940 | 0.830 |
| frac $\epsilon_r < 1\%$ *no closest material* | 0.027 | 0.077 | 0.257 | 0.497 | 0.527 |
| frac $\epsilon_r < 5\%$ *no closest material* | 0.123 | 0.550 | 0.983 | 0.862 | 0.810 |
| frac $\epsilon_r < 1\%$ | 0.012 | 0.030 | 0.147 | 0.100 | 0.055 |
| frac $\epsilon_r < 5\%$ | 0.030 | 0.187 | 0.575 | 0.498 | 0.255 |

Table 4: Further analysis of generated samples. Guidance for $J_1$ on the 2D problem with our main settings, 200 generated samples, averaged over 3 model training seeds. Results in the last two rows are identical to Table 1.

Secondly, we use the nearest neighbor distance to existing material $d_m$: In the backprojection, once the material parameters are fitted by the GMMs and correspondences are established, we look up the nearest neighbors of the two materials in our material list. The sum of the distances to these neighbors in the normalized space constitutes our metric $d_m$. Note that this part is especially challenging, since the model needs to learn which parameter vectors $(E, \nu, \rho)$ are plausible. Compare Figure 4 for the allowed resp. plausible base materials.

**Evaluation of backprojection** In Table 4 we provide further metrics for the experiment shown in Table 1. One can see that the material fitting variance $V_m$ is only slightly increased from the unguided case (see Table 5) for target bulk moduli 86.6 and 168.5 and are more strongly increased for higher targets. The target bulk modulus 4.8 exhibits an even higher variance. Distance to existing materials $d_m$ remains relatively low for small to medium target bulk moduli and increases for higher targets.

In the following, we inspect the frac metric at several stages of the backprojection. The case "sample" refers to metrics computed over the generated sample directly ($K_s$ in the main paper). The setting "no closest material" extracts parameters from the sample by fitting the material GMM and performing skeletonization. It evaluates these parameters with averaging over random spatial distributions of particles, identically to the main metrics. However it uses the extracted material parameters as-is and does not look up the closest existing material. A small performance drop can be observed from "sample" to "no closest material" in most cases. Compared to the final metrics after material lookup, the "no closest material" metrics are much higher. Especially for large target bulk moduli, a stronger performance difference can be observed. This indicates that the model can generate a high fraction of samples that are close to the target bulk modulus, but can interpolate materials, so that the used materials are not close to available ones.

To evaluate the backprojection, we also run it on our 10k 2D training dataset and report various metrics: For distance to closest material $d_m$, the mean, 99-th percentile and maximum are 1.3e-5, 7.8e-14, 0.13. Only for one sample in the dataset, $d_m$ was larger than 1e-5, which indicates a fail in recovering the correct material. For the absolute error between predicted and actual volume fraction, the mean, 99-th percentile and maximum are 1.3e-3, 0 and 0.5, respectively. Only in 42 out of the 10000 samples, the volume fraction was determined with non-zero error. For the absolute error between predicted and actual radius of the particles, the mean, 99-th percentile and maximum are 6.6e-3, 1.5e-2, 0.2. For reference, for dataset generation, we sample radii uniformly between 0.075 and 0.175. These results show that the backprojection can produce incorrect results on the clean training data, but does so only in very few cases. Also, the circle radius, which is difficult to determine, is estimated with low error.

## D   Model Architecture Details

We implement our approach in Pytorch (Ansel et al., 2024). Our diffusion model implementation is build on the `Unet2DModel` from the Diffusers library (von Platen et al., 2022). In the following, we detail the model architecture. If a setting is not specified, the default from the Diffusers library is assumed. First, the 3-channel input (normalized $E, \nu, \rho$) is embedded by a convolutional layer with kernel size 1 into 8 channels. The current timestep, varying between 0 and 999, is embedded to 16 channels with a Gaussian Fourier embedding (Tancik et al., 2020). It is then processed by a 2-layer MLP with SILU activation function to 32 channels.

The two down-blocks have output sizes 32 and 64. They process the input with two ResNet layers each, where their output sizes are equal to the whole block's output size. The ResNet layers use a kernel size of 3 for each existing spatial dimension and employ a group normalization with constant number of groups 8. They use `swish` as nonlinearity. First, the group normalization and the nonlinearity is applied, followed by the first convolution. The timestep embedding is linearly projected to the output size and added to the hidden states, after which a second group normalization is applied. This is followed again by the nonlinearity and the second convolutional layer. Both convolutional layers use the same output size. Finally, this processed result is added to the input (residual connection). To achieve this, the input is first mapped by a convolution with kernel size 1 to match the output size. After both ResNet layers, the output is downsampled with an average pooling and kernel size and stride two for existing spatial dimensions. Due to the two down-blocks used, this results in a reduction of factor 4 for the spatial dimensions in the middle of the model.

After the down-blocks, the data is processed by a mid-block which features three ResNet layers with an attention layer between each. The hidden sizes are 128 and 128 and the last layer maps back to 64, as in the input to the mid-block. The attention layers use 16 heads and a 3-dimensional positional encoding [3].

The up-blocks are build similarly and symmetrically to the down-blocks, only that they also take the output of the respective down-block as additional input ("skip connection"). The current result is concatenated with the output of the respective down-block and fed as input to each ResNet Layer (meaning, a ResNet layer by itself, which again consists of two convolutional layers). Before the input is passed to the respective ResNet layers, it is upsampled by 2 for each existing spatial dimension with `nearest` mode of `torch.nn.functional.interpolate`.

As last step in the model, the output of original spatial size and embedded channel size 8 is processed by a convolutional layer with kernel size 1 to project back to the original 3 data channels.

## E   Hyperparameters

In this section, we detail our model training and metrics which we used to optimize hyperparameters. For a definition of additional metrics used here, refer to Section C.

**Dataset size for unconditional generation**   We train our model as described in the main paper for different dataset sizes of the 2D 64x64 problem. After training, we obtain 1000 samples with 1000 diffusion steps each and compute the mean metrics. These results are shown in Table 5. We see that all metrics are relatively similar between the considered dataset sizes, with a consistent tendency of slightly lower $d_m$ for smaller sizes.

**Training and diffusion hyperparameters**   We obtained the specified hyperparameters by an empirical search. Initial experiments were conducted on models trained with a single seed on a $32 \times 32$ problem and then parameter choices were refined on models trained with three seeds on a $64 \times 64$ problem. From each trained model, 1000 samples were generated (without guidance) and evaluated according to the metrics specified above. Starting from default values provided by the framework, we iteratively searched by varying likely related parameters (e.g. $\beta_0, \beta_T$ together with the type of $\beta$-schedule) and checking whether it improved upon our previous results. If there was no considerable improvement, we kept the previous parameters. We

---

[3] https://github.com/tatp22/multidim-positional-encoding

Table 5: Comparison of different training dataset sizes for the 2D problem. 3 models with different seeds are trained for each row and 1000 samples generated each, averaged. Generation is unguided with 1000 diffusion steps.

| ds size | $V_m \downarrow$ | $d_m \downarrow$ | cov $\uparrow$ |
|---------|--------|--------|--------|
| 1k | 0.0007 | 0.1074 | 0.9742 |
| 2.5k | 0.0007 | 0.1137 | 0.9802 |
| 5k | 0.0013 | 0.1162 | 0.9841 |
| 10k | 0.0011 | 0.1164 | 0.9802 |

tried out constant and polynomial (exponent 0.5) learning rate schedules with different learning rates. For $\beta$-schedules, we tried out squared cosine and sigmoid schedules with varying $\beta_0$ and $\beta_T$. We also found the $\beta$-rescaling to be quite important for sample diversity according to our cov metric. As prediction targets, we compared predicting the noise $\epsilon$, the clean sample $x_0$ and the velocity $v$ and found that $\epsilon$-prediction performed the worst and $v$ prediction the best. We found larger batch sizes to perform similar to the baseline of 128. Regarding training step sizes, we found diminishing improvements increasing the training steps between 50k, 100k and 200k steps.

**Guidance parameters** We use the cov of $\epsilon_r < 5\%$ metric for tuning of the guidance parameters, averaged over the previously introduced subset of targets $K^*$. Guidance with DPS leaves the following parameters: The constant factor of the DPS gradient $\rho_D$ (not to be confused with a density $\rho$), the number of denoising time steps $N$ and the choice of gradient clipping schemes.

We performed initial experiments with different means of gradient normalization. In preliminary experiments, we tested several gradient normalization and clipping schemes, for example normalizing by the distance to goal as proposed by Chung et al. (2023), clipping individual components or clipping the magnitude of the gradients. We also tested scaling by the predicted noise as proposed by Shen et al. (2024). We found clipping the magnitude of gradients to be beneficial and clipping $\hat{x}_0$ per channel to restrict it to the allowed space. We then performed a grid search over $\rho_D$ and the maximum allowed gradient magnitude. We plot this experiment in Figure 9. We observe that the combination of both hyperparameters has a strong effect on performance as measured by cov. We therefore suspect that for a different problem or very different data properties, these parameters would need to be re-tuned to achieve good performance. Regarding number of guidance steps, we provide a comparison with $N = 50$ and 200 in the main paper.

**Effect of gradient scaling factor** We plot the effect of the DPS gradient scaling factor $\rho_D$ aggregated over all target bulk moduli in Figure 10. A smaller factor approaches the unconditional generation case with high sample diversity ("all" case for frac and cov metrics) but low adherence to target (low metrics in error margins). Increasing the factor generally reduces overall sample diversity while target adherence is maximized at $\rho_D = 1$. Increasing the factor beyond this point reduces all metrics again, presumably because too extreme gradients outweigh the regularization by the diffusion model.

## F   Details of Alternative Method Experiments

**Bayesian optimization** We implement this method with the framework of Nogueira (2014). A Gaussian process is fit to evaluated data points and their objective value. The next data point to evaluate is determined via an upper confidence bound (UCB) formulation. Concretely, the term $\mu + \kappa \sigma$ is sought to be maximized. We conducted several hand-crafted experiments and found $\kappa = 1$ to yield best results in terms of our cov metric in the $\epsilon_r < 5\%$ interval. Notably, finding the next data point to evaluate requires solving (even if not to optimality) an optimization problem, which takes significant time. This time is increased with the number of obtained data points. For our experiments, we found that between 750 and 1000 iterations (albeit more data points are added; see below) it takes roughly 10 seconds to suggest a new data point.

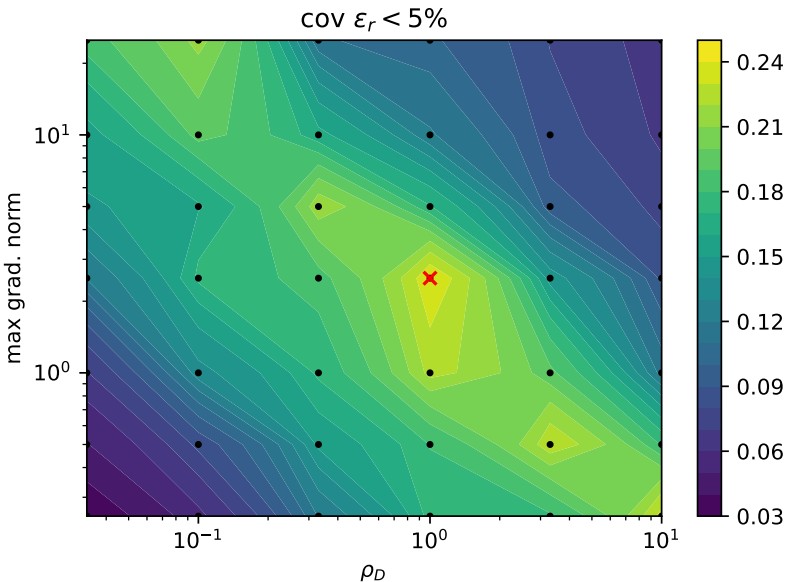

Figure 9: Effect of parameters $\rho_D$ and maximum gradient magnitude on $J_1$. Computed over 200 samples, averaged over 3 target bulk moduli. Black dots show evaluated data points, the red cross indicates the value used in our main experiments.

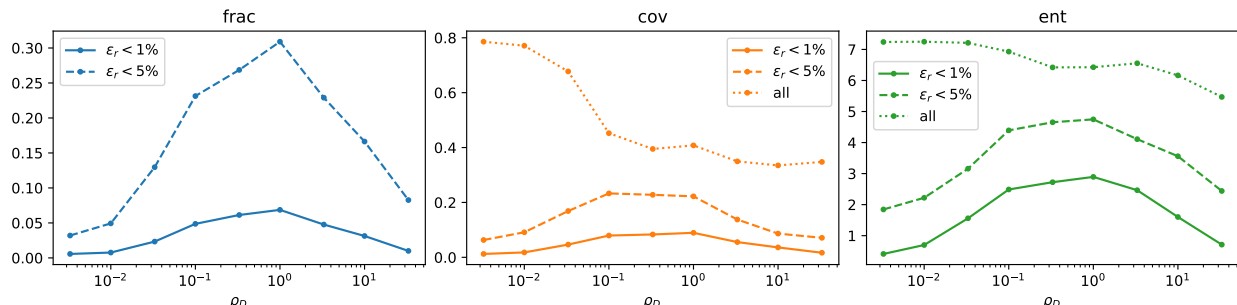

Figure 10: Effect of parameter $\rho_D$ on all metrics. Mean metrics over all targets $K^*$ for 200 samples each and over 3 model training seeds. The values at $\rho_D = 1$ are identical to Table 2a.

We represent the parameter space $\Theta$ continuously as subset of $\mathbb{R}^8$. Since only several discrete material parameters are possible, we project the data point suggested by the acquisition function to closest existing materials, similarly to our backprojection. We then evaluate this parameter with random sampling of microstructures, identically to the evaluation of our method (compare Section 6.2) and assign the resulting value to both the suggested and actually evaluated data point. Note that the former is required to reduce the variance at the suggested point so that it will not be suggested again. Formally, the continuous space of material properties of matrix and particles can be represented as a Voronoi diagram, where the given set of points are the available materials. Each suggested point in a Voronoi region will be projected to the closest existing material and therefore has identical cost value. We experimented with sampling the Voronoi region after the first point in it is evaluated to add multiple data points to the Gaussian process, but found that the increased computation time for suggestion outweighs the potentially redundant evaluations. To obtain a parameter for evaluation, a value is first randomly sampled from the space and used as initial value for the maximization of $\mu$ (without an exploration term).

**Reinforcement Learning**   We use the soft actor-critic method (Haarnoja et al., 2018) for our RL experiments as it has two advantages over other RL methods: It is an off-policy method that re-uses previous interactions, which is important in a scenario like ours where evaluating an action (design proposal) is expensive. Secondly, the formulation allows to provide a target entropy to enforce some stochasticity over actions. This is especially important as we are interested in diverse designs. A deterministic agent would converge onto the same design for a target. We consider a similar setting as Würz & Weißenfels (2025), where the agent interacts in multiple steps with the environment, where an action corresponds to proposed design parameters. The environment determines closest existing materials and evaluates the bulk modulus over random spatial arrangements. During training, we only use a single spatial arrangement similarly to Würz & Weißenfels (2025) and for performance evaluation 10 spatial arrangements, similarly to the evaluations for other methods in this paper. As observation, the actually used materials as well as signed relative and absolute error to the target bulk modulus are returned to the agent. We use 10 steps per episode. To compute an initial observation, fully random parameters are sampled and evaluated to provide the agent with a starting point.

We use the reward formulation from (Würz & Weißenfels, 2025) and use their critic network architecture for both our actor and critic networks. We use the implementation provided by Raffin et al. (2021). Similarly to BO, we tune the following hyperparameters on the three validation bulk modulus targets on the cov $\epsilon_r < 5\%$ metric: number of gradient steps per environment interaction $n_g$, target entropy and discount factor $\gamma$. We leave other hyperparameters at their default values from the implementation of Raffin et al. (2021). We determine hyperparameters in the following way: First, we determine the best value of $n_g$ at 10k environment interactions as $n_g = 16$. We then tune the target entropy. We observe that for the default value (-8), cov $\epsilon < 5\%$ continually decreases after 5k env steps until about 50k steps while the frac metrics rise. This means that the agent converges to few design parameters that bring highest reward. We observe that indeed the target entropy parameter serves as a tradeoff between frac and cov or between a high reward for approaching the objective and the diversity of proposed designs. With higher target entropy, higher cov metrics can be achieved for ongoing training. We determine the target entropy at 30k env steps, where we assume that the values will not drop significantly further during training. We determine the best value as target entropy 4. With this value, we observe no relevant drop in cov after 10k env steps. We determine further hyperparameters at 15k environment steps. We tested activation functions `relu` and `tanh` instead of the `gelu` used in Würz & Weißenfels (2025), but find the latter to perform best. We then determine the best discount factor as $\gamma = 0.9$. Finally, we check whether a different $n_g$ increases cov performance with the currently determined parameters, but find that $n_g = 16$ still performs best.

We run training for 95k env steps and observe that metrics do not significantly improve after 35k steps. We provide results at this step for 3 training seeds.

**Conditionally trained diffusion models**   Firstly, we compute the bulk modulus $K$ on all examples in the training set. To embed the bulk modulus, we first map the interval of occurring values to the interval $[0, 1]$. We then embed it similarly to the diffusion timestep and after processing with a dedicated MLP, both embeddings are added before being input to the convolutional layers. Following Ho & Salimans (2022), we trained models with a probability to replace the conditional input by a null embedding $p \in \{0, 0.1, 0.2\}$. For the models with non-zero probability, we tried out several values for the guidance scale $w$ and found that the combination of $p = 0.1$, $w = 0.5$ performed best (using the scale formulation $w$ as in the cited paper). In the main paper, we report results obtained with that parameter combination. Apart from that, we use identical settings for the diffusion model and sampling as for our approach (but not using any gradient guidance).

## G   Additional Results

**Qualitative results of remaining targets**   We show the visualizations of remaining targets $K^*$ in Figure 11 and Figure 12.

**Runtime analysis**   We investigate execution times of our approach further in Table 6. Results are reported for a cluster node using 16 CPU cores and an A40 GPU. Three instances of the FEM solver run in parallel

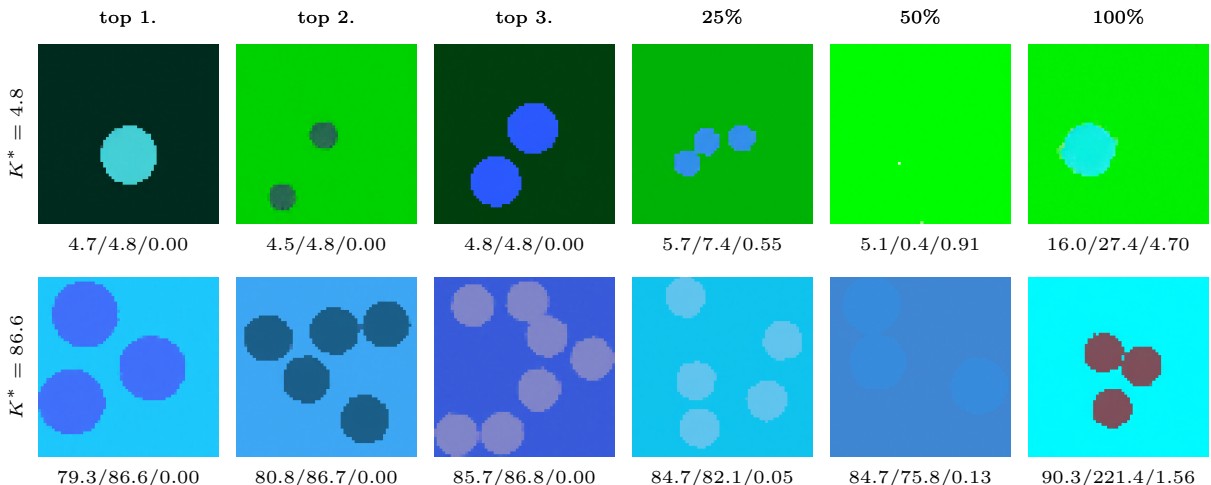

Figure 11: Inverse 2D material designs. Generated samples for selected bulk moduli $K^*$, ordered by relative error quantile. Best on the left, worst on the right. Labels show $K_s$ / $K_\theta$ / $\epsilon_r$. The values $(E, \nu, \rho)$ in the normalized coordinate space are encoded as $(r, g, b)$ values of the image.

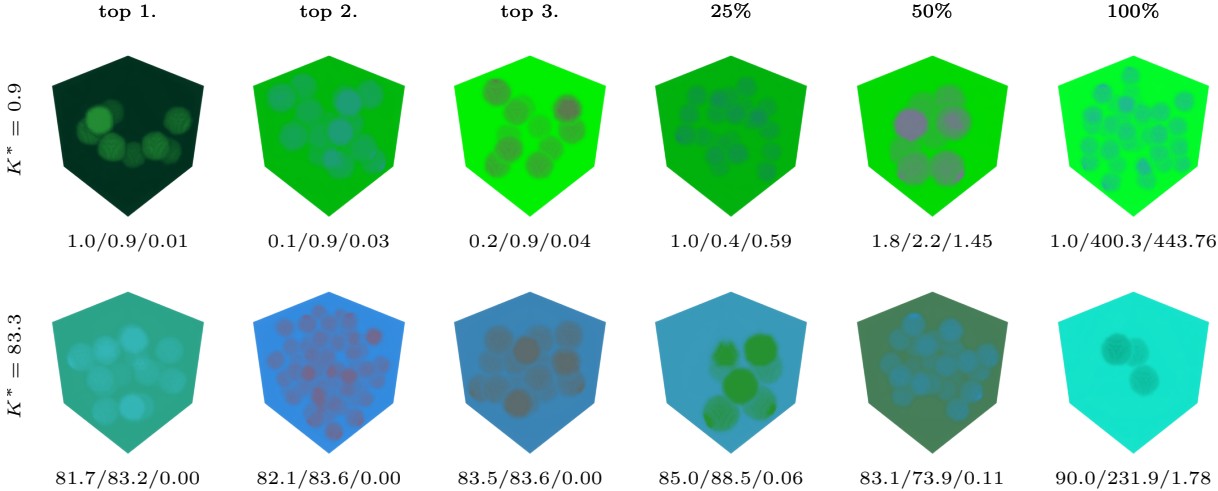

Figure 12: Inverse 3D material designs. Generated samples for selected bulk moduli $K^*$, ordered by relative error quantile. Best on the left, worst on the right. Labels show $K_s$ / $K_\theta$ / $\epsilon_r$. The values $(E, \nu, \rho)$ in the normalized coordinate space are encoded as $(r, g, b)$ values of the image.

on CPU and each instance uses multiple threads. The dominating factor of the pipeline is the solution of the FEM problem and the computation of FEM gradients in the analytical solver ("FEM solve & diff.").

Table 6: Runtime analysis of our approach. Results are reported for the generation of one batch of the specified size for $J_1$, averaged over 5 target bulk moduli, 200 samples total. Guided diffusion with 100 steps (99 gradient computations per sample). The FEM gradient computation clearly dominates the runtime.

| setting | batch size | total time | FEM solve & diff. (fraction) |
|---|---|---|---|
| 2D 32x32 | 50 | 387 s | 385 s (99.4%) |
| 2D 64x64 | 50 | 1241 s | 1236 s (99.5%) |

Table 7: Standard deviations of Table 1 (comparison of our method, projection variant and alternative inverse design approaches on the 2D problem on objective $J_1$), computed over 3 training seeds for each method.

| $K^*$ | | frac | | | | | cov | | | | | ent | | | | |
| | | 4.8′ | 86.6 | 168.5′ | 250.4 | 332.2′ | 4.8′ | 86.6 | 168.5′ | 250.4 | 332.2′ | 4.8′ | 86.6 | 168.5′ | 250.4 | 332.2′ |
|---|---|---|---|---|---|---|---|---|---|---|---|---|---|---|---|---|
| ours | $\epsilon_r < 1\%$ | 0.008 | 0.028 | 0.013 | 0.035 | 0.017 | 0.015 | 0.053 | 0.009 | 0.038 | 0.015 | 1.000 | 1.765 | 0.123 | 0.212 | 0.304 |
| | $\epsilon_r < 5\%$ | 0.013 | 0.058 | 0.013 | 0.075 | 0.092 | 0.019 | 0.036 | 0.009 | 0.095 | 0.040 | 0.664 | 0.385 | 0.010 | 0.328 | 0.426 |
| ours, proj. | $\epsilon_r < 1\%$ | 0.003 | 0.019 | 0.034 | 0.049 | 0.025 | 0.006 | 0.028 | 0.015 | 0.042 | 0.024 | 0.577 | 0.375 | 0.175 | 0.424 | 0.987 |
| | $\epsilon_r < 5\%$ | 0.006 | 0.066 | 0.026 | 0.077 | 0.111 | 0.023 | 0.019 | 0.038 | 0.074 | 0.068 | 0.371 | 0.039 | 0.040 | 0.536 | 0.520 |
| cond. diff | $\epsilon_r < 1\%$ | 0.006 | 0.010 | 0.030 | 0.035 | 0.028 | 0.014 | 0.012 | 0.042 | 0.034 | 0.025 | 0.577 | 0.255 | 0.295 | 0.407 | 0.742 |
| | $\epsilon_r < 5\%$ | 0.006 | 0.035 | 0.036 | 0.036 | 0.005 | 0.007 | 0.023 | 0.033 | 0.003 | 0.016 | 0.194 | 0.130 | 0.077 | 0.043 | 0.046 |
| BO | $\epsilon_r < 1\%$ | 0.003 | 0.036 | 0.033 | 0.064 | 0.049 | 0.007 | 0.006 | 0.052 | 0.060 | 0.037 | 0.000 | 0.392 | 0.655 | 0.859 | 0.424 |
| | $\epsilon_r < 5\%$ | 0.003 | 0.113 | 0.137 | 0.227 | 0.119 | 0.007 | 0.024 | 0.115 | 0.080 | 0.086 | 0.000 | 0.208 | 0.651 | 0.514 | 0.257 |
| RL | $\epsilon_r < 1\%$ | 0.000 | 0.018 | 0.015 | 0.024 | 0.006 | 0.000 | 0.027 | 0.007 | 0.012 | 0.012 | 0.000 | 0.269 | 0.090 | 0.101 | 0.105 |
| | $\epsilon_r < 5\%$ | 0.003 | 0.019 | 0.015 | 0.035 | 0.006 | 0.007 | 0.016 | 0.014 | 0.024 | 0.012 | 0.000 | 0.103 | 0.065 | 0.120 | 0.079 |
| ours | $\epsilon < 1$ | 0.010 | 0.026 | 0.015 | 0.008 | 0.003 | 0.015 | 0.048 | 0.025 | 0.026 | 0.000 | 0.112 | 1.233 | 0.267 | 0.369 | 0.230 |
| | $\epsilon < 5$ | 0.118 | 0.051 | 0.016 | 0.053 | 0.033 | 0.030 | 0.027 | 0.024 | 0.051 | 0.025 | 0.497 | 0.284 | 0.035 | 0.153 | 0.326 |
| | $\epsilon < 10$ | 0.066 | 0.038 | 0.030 | 0.079 | 0.052 | 0.036 | 0.043 | 0.018 | 0.083 | 0.027 | 0.407 | 0.094 | 0.036 | 0.303 | 0.413 |

Table 8: Standard deviations of Table 2 (ablations and guidance with $J_2$), computed over 3 model training seeds and all 5 target bulk moduli.

(a) Performance for guidance by $J_1$, variants.

| | $\epsilon_r < 1\%$ | | | $\epsilon_r < 5\%$ | | |
| | frac | cov | ent | frac | cov | ent |
|---|---|---|---|---|---|---|
| ds 10k | 0.054 | 0.080 | 1.567 | 0.214 | 0.159 | 1.426 |
| ds 1k | 0.050 | 0.075 | 1.637 | 0.238 | 0.150 | 2.138 |
| $N = 50$ | 0.048 | 0.067 | 1.610 | 0.191 | 0.173 | 1.340 |
| $N = 200$ | 0.057 | 0.051 | 1.414 | 0.215 | 0.131 | 1.782 |
| unguided | 0.003 | 0.007 | 0.258 | 0.021 | 0.040 | 1.450 |
| project | 0.105 | 0.101 | 1.777 | 0.316 | 0.153 | 1.431 |

(b) Performance for guidance by $J_2$ (also minimize density). Additionally we show the average density of samples that fall in the respective error margin.

| | $\epsilon_r < 1\%$ | | | | $\epsilon_r < 5\%$ | | | |
| | $\rho_{\mathrm{avg}}$ | frac | cov | ent | $\rho_{\mathrm{avg}}$ | frac | cov | ent |
|---|---|---|---|---|---|---|---|---|
| $\lambda = 0$ | 2.612 | 0.054 | 0.080 | 1.567 | 2.390 | 0.214 | 0.159 | 1.426 |
| $\lambda = 10^{-4}$ | 2.154 | 0.056 | 0.072 | 1.370 | 2.272 | 0.230 | 0.167 | 1.225 |
| $\lambda = 10^{-3}$ | 1.133 | 0.035 | 0.051 | 1.501 | 1.448 | 0.195 | 0.146 | 2.069 |
| $\lambda = 10^{-2}$ | 0.919 | 0.020 | 0.021 | 1.284 | 1.034 | 0.106 | 0.046 | 1.999 |

**Standard deviations of tables** We provide the standard deviations of Table 1 in Table 7. One an see that for our approach, the variance over model training seeds is relatively low at targets with high performance, e.g. $K^* = 168$. This indicates that the method achieves consistently good performance for these targets under model training variability. Targets with low performance, such as $K^* = 4.8$ exhibit considerable variance over model training seeds, which is expected since fewer samples fall into the respective margins for metric computation. Standard deviations of Table 2 are shown in Table 8. They are generally larger as the per-target ones, which likely stems from the varying performance over material targets (as can be seen in Table 1). The conditional diffusion model exhibits low variance over trained model seeds, in a similar range as our guidance results, except that some $\epsilon_r < 5\%$ metrics exhibit much lower variances (see Table 7). The Bayesian optimization approach exhibits a much higher variance over initial random states than our method in most cases for $K^* = 168.5$ and higher targets. The RL approach achieves lower variances over training seeds than our approach in most cases. Standard deviations of Table 3 (3D problem) are presented in Table 9 and lie in a similar range as the 2D problem.

Table 9: Standard deviations of Table 3 (evaluation of our method on the 3D problem), computed over 3 training seeds.

| | metric: frac | | | | | cov | | | | | ent | | | | |
|---|---|---|---|---|---|---|---|---|---|---|---|---|---|---|---|
| $K^*$: | 0.9 | 83.3 | 165.6 | 248.0 | 330.3 | 0.9 | 83.3 | 165.6 | 248.0 | 330.3 | 0.9 | 83.3 | 165.6 | 248.0 | 330.3 |
| $\epsilon_r < 1\%$ | 0.003 | 0.009 | 0.008 | 0.060 | 0.037 | 0.007 | 0.012 | 0.016 | 0.009 | 0.021 | 0.000 | 0.263 | 0.117 | 0.122 | 1.639 |
| $\epsilon_r < 5\%$ | 0.005 | 0.044 | 0.036 | 0.083 | 0.160 | 0.009 | 0.018 | 0.024 | 0.027 | 0.052 | 0.338 | 0.265 | 0.071 | 0.079 | 1.105 |
| $\epsilon < 1$ | 0.045 | 0.018 | 0.010 | 0.029 | 0.013 | 0.040 | 0.027 | 0.009 | 0.018 | 0.016 | 0.005 | 0.423 | 0.245 | 0.403 | 0.923 |
| $\epsilon < 5$ | 0.053 | 0.051 | 0.042 | 0.103 | 0.064 | 0.042 | 0.012 | 0.027 | 0.019 | 0.028 | 0.029 | 0.248 | 0.134 | 0.168 | 1.006 |
| $\epsilon < 10$ | 0.010 | 0.043 | 0.057 | 0.119 | 0.125 | 0.045 | 0.018 | 0.027 | 0.010 | 0.043 | 0.096 | 0.127 | 0.115 | 0.052 | 1.133 |

