# OpenReview forum: "Guided Diffusion by Optimized Loss Functions on Relaxed Parameters for Inverse Material Design"
_TMLR — Decision pending for TMLR_

### Review · Reviewer_4ZWp · 2026-03-26

**Summary Of Contributions:**

This work proposes to solve the problem of inverse material design relying on guided diffusion methods to encourage more diverse solutions. To be able to use diffusion models, the authors relax the parameter space to be able to compute gradient thanks to implicit differentiation. Then, they train an unconditional diffusion model, before using loss guidance. Then, they experiment on inverse material designs, perform ablations, and compare with baselines.

**Audience:**

Yes

**Audience Explanation:**

While the application is in inverse material design, the methodological contribution (combining diffusion models with gradients obtained via implicit differentiation through simulators) is broadly relevant to the ML community working on inverse problems and physics-informed learning.

**Claims And Evidence:**

Yes

**Claims Explanation:**

The claims made in the submission seem accurate.

**Requested Changes:**

Overall, the paper is well written, and the results seem sound. The method proposed seems to solve problems of previous inverse material design methods, and it has been heavily tested in the experimental section.

Nonetheless, I believe that the paper in the current state may be hard to follow for Machine Learning audience, which is the one of TMLR. I think the paper would be clearer with a better introduction and more background on inverse material design. For instance, words such as "bulk modulus" often appear, but I don't know what it is. Additionally, the presentation of diffusion models lacks clarity. For instance, it is said that the velocity parametrization is used (in Section 3.1), but it is not clear how it relates to the formula introduced before.

More importantly, several methodological components would benefit from deeper explanation. The relaxation of the parameter space, the role of the diffusion model as a prior, and the backprojection step are central to the approach but are not sufficiently motivated or analyzed. In particular, the backprojection relies on heuristic procedures,which may introduce errors and deserve further discussion.


**Typos**:
- Initials FEM used well before the introduction of the full words (Section 5.1)
- Initials DDIM not introduced.
- Citation Zampini et al: "NeurIPS 2025. To appear"
- Page 5: weird sentende: "We consider an inverse material design problem in which circular respectively spherical particles of a specific material, radius, and volume fraction are mixed into a matrix made of another material."

---

> ### Author Response · Authors · 2026-05-31
> **Answer to Reviewer 4ZWp**
>
> We thank the reviewer for their time and constructive comments.
> We have revised the paper accordingly and respond to the reviewer's comments below:
> > I believe that the paper in the current state may be hard to follow for Machine Learning audience, which is the one of TMLR. I think the paper would be clearer with a better introduction and more background on inverse material design. For instance, words such as "bulk modulus" often appear, but I don't know what it is.
> Additionally, the presentation of diffusion models lacks clarity. For instance, it is said that the velocity parametrization is used (in Section 3.1), but it is not clear how it relates to the formula introduced before.
>
> > More importantly, several methodological components would benefit from deeper explanation. The relaxation of the parameter space, the role of the diffusion model as a prior, and the backprojection step are central to the approach but are not sufficiently motivated or analyzed.
> In particular, the backprojection relies on heuristic procedures,which may introduce errors and deserve further discussion.
>
> - We have added more background on inverse material design to the introduction and hope that it is now easier to follow for a general machine learning audience.
> We have also clarified the use of the $\mathbf{v}$ parameterization for the diffusion model.
>
> - Diffusion models learn an approximation of the data distribution. This can be used to learn the distribution over design parametrizations. At inference time, loss guidance allows for steering the samples towards objective functions. The objective function can be flexibly chosen during inference time. However, it needs to be differentiable for the sample space of the diffusion model. In our approach, we propose to relax design parameters to make the objective function based on FEM differentiable. To yield valid design parameters, a backprojection is needed. This motivation is detailed in Sec. 1.
>
> - We perform a deeper analysis of the backprojection in appendix section C.
> We evaluate potential errors of the backprojection process on our training data set.
> We also perform a detailed analysis of errors of generated samples at different stages of the backprojection. We have added a reference to this chapter to the results discussion in the main paper.

---

### Review · Reviewer_6Q7Z · 2026-04-27

**Summary Of Contributions:**

This paper proposes a diffusion-based pipeline for matrix-particle composite inverse design. It trains an unconditional diffusion prior on a continuous FEM grid, guides sampling using FEM gradients obtained via implicit differentiation, and projects relaxed samples back to discrete material parameters. Experiments evaluate 2D/3D bulk modulus matching and a density-regularized objective.

## Strengths
The problem is well motivated, and the idea of combining a generative prior with simulator-based guidance is reasonable. The main potential strength is objective flexibility: unlike conditional diffusion, the method can in principle change objectives at inference time without retraining the generative model.

## Weaknesses
However, the manuscript currently over-claims its contribution. Diffusion-based inverse design, loss-guided diffusion, and differentiable FEM have all been explored in prior work. The paper therefore needs to more clearly articulate what is substantively new beyond combining existing components.

**Audience:**

Yes

**Audience Explanation:**

The paper should be of interest to readers working on diffusion models for inverse problems, physics-guided generative modeling, differentiable simulation, and computational material design.

**Claims And Evidence:**

No

**Claims Explanation:**

1. **The contribution is over-claimed, and the distinction from prior work is not systematically discussed.**

   The paper claims to propose a novel approach for inverse design, and presents loss-guided diffusion, implicit differentiation, and material-design evaluation as key contributions. However, these components have substantial overlap with existing work: diffusion-based material/metamaterial inverse design has already been studied [1,2], loss-guided or training-free diffusion has been developed in more general settings [3], and differentiable FEM-based inverse design is also not new. Although the Related Work section mentions these relevant papers, the manuscript does not sufficiently clarify what is substantively new beyond existing work.

2. **The experimental results do not fully support the strong claim of “closely and diversely” achieving target properties.**

   The paper claims that the method can generate diverse material designs that closely achieve a wide range of medium-to-high target bulk moduli. However, under the 1% relative error threshold, the success rate is often only a few percent to around 10%.

3. **The conditional diffusion baseline weakens the empirical case for the proposed method.**

   On the main target-matching task, the conditional diffusion baseline is highly competitive and outperforms the proposed method in some settings. The authors therefore need to more directly demonstrate the practical advantage of the proposed method over conditional diffusion.

4. **Backprojection is an important step, but it is insufficiently analyzed.**

   The final evaluation is performed on the backprojected design, not directly on the relaxed sample generated by diffusion. This backprojection involves material clustering, geometry extraction, radius estimation, volume-fraction estimation, and nearest-neighbor material matching, each of which may introduce substantial error. The paper should separate errors due to diffusion guidance from errors introduced by the backprojection pipeline.



[1] Y. Yang, L. Wang, X. Zhai, K. Chen, W. Wu, Y. Zhao, L. Liu, and X.-M. Fu. *Guided Diffusion for Fast Inverse Design of Density-based Mechanical Metamaterials*. arXiv:2401.13570, 2024.

[2] J.H. Bastek and D. M. Kochmann. *Inverse design of nonlinear mechanical metamaterials via video denoising diffusion models*. *Nature Machine Intelligence*, 2023.

[3] S. Zampini, J. K. Christopher, L. Oneto, D. Anguita, and F. Fioretto. *Training-Free Constrained Generation With Stable Diffusion Models*. NeurIPS, 2025.

**Requested Changes:**

1. **Clarify and narrow the contribution**

   The authors should explicitly position the paper against prior work on diffusion-based inverse design, loss/training-free diffusion guidance, and differentiable FEM inverse design.

2. **Moderate performance claims.**

3. **Analyze backprojection error.**

   The authors should report errors before and after backprojection and identify how much error comes from material matching, geometry extraction, and radius/volume-fraction estimation.

4. **Add necessary experiments to clearly demonstrate the advantage of the proposed method.**

---

> ### Author Response · Authors · 2026-05-31
> **Answer to Reviewer 6Q7Z, part 1**
>
> We thank the reviewer for their time to review our paper and providing valuable suggestions.
> We have revised our paper accordingly.
> Below we detail how we addressed them:
> > **The contribution is over-claimed, and the distinction from prior work is not systematically discussed.**
> The paper claims to propose a novel approach for inverse design, and presents loss-guided diffusion, implicit differentiation, and material-design evaluation as key contributions. However, these components have substantial overlap with existing work: diffusion-based material/metamaterial inverse design has already been studied [1,2], loss-guided or training-free diffusion has been developed in more general settings [3], and differentiable FEM-based inverse design is also not new. Although the Related Work section mentions these relevant papers, the manuscript does not sufficiently clarify what is substantively new beyond existing work.
>
> > **Clarify and narrow the contribution**
>     The authors should explicitly position the paper against prior work on diffusion-based inverse design, loss/training-free diffusion guidance, and differentiable FEM inverse design.
>
> Thanks for pointing this out. We have toned down claims and reformulated our contribution in section 1. We have also added an explicit statement regarding parts we use in our method that have been proposed in previous works and highlighted our focus on developing a machine learning approach for inverse design that relaxes the parameter space and combines the parts in a novel way.
> We discuss similarities and differences to previous works in this regard in our related work section. We have added further explanation of differences to DPS and DRL methods.
>
>
> > **The experimental results do not fully support the strong claim of “closely and diversely” achieving target properties.**
> The paper claims that the method can generate diverse material designs that closely achieve a wide range of medium-to-high target bulk moduli. However, under the 1% relative error threshold, the success rate is often only a few percent to around 10%.
>
> > **Moderate performance claims.**
>
> We have revised the wording, e.g., in abstract, Sec. 1, and conclusions. E.g. Sec. 1:
> "Our results demonstrate that our approach can propose multiple diverse materials within a 5\% relative error margin for all target bulk moduli and within 1\% relative error margin for medium to high target bulk moduli."
>
>
> > **The conditional diffusion baseline weakens the empirical case for the proposed method.**
>   On the main target-matching task, the conditional diffusion baseline is highly competitive and outperforms the proposed method in some settings. The authors therefore need to more directly demonstrate the practical advantage of the proposed method over conditional diffusion.
>
> > **Add necessary experiments to clearly demonstrate the advantage of the proposed method.**
>
> The main advantage of our approach compared to guided diffusion is its flexibility concerning the objective function.
> As we note in section 6.3, an objective function like our $J_2$ which penalizes density without an explicit density target cannot be implemented with conditional diffusion without further means.
> Also, for new types of conditionals, the conditional diffusion model needs to be retrained while our approach can be adapted to various objective functions in a zero-shot way, as we demonstrate with the experiments that also minimize sample density.
> This might be even more relavant on larger or more complex datasets where re-training a model might be prohibitive.
>
> In the presented comparison on $J_1$, we do not expect to outperform conditional diffusion, since it has an advantage by training on the objective, and view it as a point of reference.
> Note that the diffusion model we use is agnostic of the bulk modulus whereas the conditional diffusion model is explicitly trained with the bulk modulus as input.
> Notably, for our updated results due to changes in guidance parameters (see general comment), our model achieves comparable or better performance than the conditional diffusion model in several cases.

---

> ### Author Response · Authors · 2026-05-31
> **Answer to Reviewer 6Q7Z, part 2**
>
> > **Backprojection is an important step, but it is insufficiently analyzed.**
>   The final evaluation is performed on the backprojected design, not directly on the relaxed sample generated by diffusion. This backprojection involves material clustering, geometry extraction, radius estimation, volume-fraction estimation, and nearest-neighbor material matching, each of which may introduce substantial error. The paper should separate errors due to diffusion guidance from errors introduced by the backprojection pipeline.
>
> > **Analyze backprojection error.**
>     The authors should report errors before and after backprojection and identify how much error comes from material matching, geometry extraction, and radius/volume-fraction estimation.
>
> Please refer to appendix section C for a detailed analysis of errors at different stages of the backprojection. We added a reference to this chapter to the results discussion in the main paper.
> We find that the largest performance degradation, especially for high target bulk moduli, comes from material matching, which indicates that the diffusion model interpolates materials.

---

### Review · Reviewer_uFiX · 2026-05-18

**Summary Of Contributions:**

This paper proposes a guided diffusion approach for inverse material design. The key idea is to relax a partially discrete design parameter space into a continuous grid representation (pixel/voxel), train an unconditional diffusion model as a prior over valid microstructures, and guide sampling at inference time using gradients propagated through a differentiable FEM solver via implicit differentiation. A domain-specific backprojection step maps the generated continuous sample back to the original discrete design space. The method is evaluated on a composite material design task (targeting a prescribed bulk modulus) in both 2D and 3D settings.

**Audience:**

Yes

**Audience Explanation:**

The intersection of diffusion models and physics-based simulation is an active and relevant research area.

**Claims And Evidence:**

No

**Claims Explanation:**

Two main concerns undermine the experimental support for the claims.

**Guidance Scaling Hyperparameters)** The method requires manually tuned, per-channel gradient scaling factors (0.5 for E, 0.02 for ν), found via grid search. No justification is provided for why these specific values are needed. It is also unclear whether the same scaling factors would transfer to a different material system or a different FEM problem. More importantly, the guidance scale directly controls the trade-off between satisfying the objective and diversity, which the authors mention as a limitation of prior approaches such as BO and RL. The paper does not include an analysis of how varying these scaling factors affects this trade-off.

**Baseline Comparison)** The paper compares against Bayesian optimization (BO) and a conditionally trained diffusion model. I believe that including RL-based baselines will support authors' claim. Moreover, I'm not sure that frac and cov measure can reflect the diversity of generated design parameters. The authors should consider complementary metrics to more convincingly support the claim that the method generates diverse designs.

**Requested Changes:**

- Provide an analysis of how guidance scaling hyperparameters affect the diversity–performance trade-off.  Discuss whether these values generalize beyond the specific material system evaluated.

- Comparison with RL-based approaches. Reorganize the baseline comparison into a single unified table that allows direct row-by-row comparison of the proposed method against BO and the conditional diffusion model across all target K* values. It is hard to compare the results.

---

> ### Author Response · Authors · 2026-05-31
> **Answer to Reviewer uFiX**
>
> We thank the reviewer for their effort and suggestions to improve our paper.
> Please see below for our response to the suggestions which we have also addressed in the paper including additional experiments.
>
> > **Guidance Scaling Hyperparameters:**
> The method requires manually tuned, per-channel gradient scaling factors (0.5 for E, 0.02 for ν), found via grid search. No justification is provided for why these specific values are needed. It is also unclear whether the same scaling factors would transfer to a different material system or a different FEM problem. More importantly, the guidance scale directly controls the trade-off between satisfying the objective and diversity, which the authors mention as a limitation of prior approaches such as BO and RL. The paper does not include an analysis of how varying these scaling factors affects this trade-off.
>
> > Provide an analysis of how guidance scaling hyperparameters affect the diversity–performance trade-off. Discuss whether these values generalize beyond the specific material system evaluated.
>
> - We agree that per-channel gradient scaling adds unnecessary heuristics and complexity. We have performed a new search over guidance hyperparameters without per-channel scaling parameters.
> We have updated the results in the paper to this setting.
> Regarding performance metrics for relative error margins, the values have generally improved for most targets and decreased for some.
> - We have added an additional evaluation of the effect of the single guidance scaling parameter $\rho_D$ to appendix section E. As anticipated, increasing it generally reduces overall sample diversity. Adherence to the objective increases with higher scaling parameter up to some point.
> - We added a discussion regarding re-tuning of guidance hyperparameters to appendix section E. Since performances is strongly impacted by their choice, we suspect that for a different problem or very different data properties, these parameters would need to be re-tuned to achieve good performance.
>
> > **Baseline Comparison)** [...] I believe that including RL-based baselines will support authors' claim. Moreover, I'm not sure that frac and cov measure can reflect the diversity of generated design parameters. The authors should consider complementary metrics to more convincingly support the claim that the method generates diverse designs.
>
> > Comparison with RL-based approaches. Reorganize the baseline comparison into a single unified table that allows direct row-by-row comparison of the proposed method against BO and the conditional diffusion model across all target K* values.
>
> - We implemented an RL baseline using the soft actor-critic method. Note that an RL policy is trained from scratch for each specific $K$ target value. Results demonstrate that the RL baseline achieves comparable or in some cases better results. However, we note that our method is more flexible, since objective as well as target values can be changed without retraining at inference time.
>
> - We have introduced a new diversity metric that measures the entropy over different discretizations of generated design parameters.
> We re-organized baseline comparisons into a single unified table.

---

### Author Response · Authors · 2026-05-31
**Answer to all reviewers**

We thank the reviewers for their time and valuable feedback.
We are glad that the reviewers find that our approach is relevant to the community (uFiX,4ZWp), the idea of combining diffusion models as generative prior with simulation for guidance is reasonable and allows for flexibility in the objective (6Q7Z), the approach has been heavily tested (4ZWp), and the paper is well written (4ZWp).
We have incorporated the feedback into the paper accordingly and provide an explanation of our responses to each suggestion below.

As a general notice, as noted by reviewer uFiX, we used tuned per-channel gradient scaling factors in our method, which is a heuristic that we reconsidered and removed from the paper.
We now report results without per-channel scaling parameters.
Hence, several result tables and figures had to be updated.
Regarding performance metrics for relative error margins, the values have generally improved for most targets and decreased for some.
Also, on request of reviewer uFiX, we have introduced a new diversity performance metric that measures the entropy over different discretizations of generated design parameters.
To this end, we have repeated some experiments.
We have highlighted changes in the paper text by blue color.